# Heparan Sulfate Proteoglycans as Potential Markers for In Vitro Human Neural Lineage Specification

**DOI:** 10.3390/cells14151158

**Published:** 2025-07-26

**Authors:** Chieh Yu, Duy L. B. Nguyen, Martina Gyimesi, Ian W. Peall, Son H. Pham, Lyn R. Griffiths, Rachel K. Okolicsanyi, Larisa M. Haupt

**Affiliations:** 1Centre for Genomics & Personalised Health, Genomics Research Centre, School of Biomedical Sciences, Queensland University of Technology, Brisbane, QLD 4059, Australia; cedricyu@stanford.edu (C.Y.); lebaoduy.nguyen@hdr.qut.edu.au (D.L.B.N.); martina.gyimesi@hdr.qut.edu.au (M.G.); ian.peall@hdr.qut.edu.au (I.W.P.); h27.pham@qut.edu.au (S.H.P.); lyn.griffiths@qut.edu.au (L.R.G.); rachelkmackay@gmail.com (R.K.O.); 2ARC Training Centre for Cell and Tissue Engineering Technologies, Queensland University of Technology (QUT), Kelvin Grove, Brisbane, QLD 4059, Australia; 3Max Planck Queensland Centre for the Materials Sciences of Extracellular Matrices, Queensland University of Technology, Brisbane, QLD 4059, Australia

**Keywords:** heparan sulfate proteoglycans, neural cell lines, human neurogenesis, syndecan, glypican, lineage differentiation

## Abstract

Heparan sulfate proteoglycans (HSPGs) within the neuronal niche are expressed during brain development, contributing to multiple aspects of neurogenesis, yet their roles in glial lineage commitment remain elusive. This study utilised three human cell models expanded under basal culture conditions followed by media-induced lineage induction to identify a reproducible and robust model of gliogenesis. SH-SY5Y human neuroblastoma cells (neuronal control), ReNcell CX human neural progenitor cells (astrocyte inductive) and ReNcell VM human neural progenitor (mixed neural induction) models were examined. The cultures were characterised during basal and inductive states via Q-PCR, Western Blotting, immunocytochemistry (ICC) and calcium signalling activity analyses. While the ReNcell lines did not produce fully mature or homogeneous astrocyte cultures, the ReNcell CX cultures most closely resembled an astrocytic phenotype with ReNcell VM cells treated with platelet-derived growth factor (PDGF) biased toward an oligodendrocyte lineage. The glycated variant of surface-bound glypican-2 (GPC2) was found to be associated with lineage commitment, with GPC6 and 6-*O* HS sulfation upregulated in astrocyte lineage cultures. Syndecan-3 (SDC3) emerged as a lineage-sensitive proteoglycan, with its cytoplasmic domain enriched in progenitor-like states and lost upon differentiation, supporting a role in maintaining neural plasticity. Conversely, the persistence of transmembrane-bound SDC3 in astrocyte cultures suggest continued involvement in extracellular signalling and proteoglycan secretion, demonstrated by increased membrane-bound HS aggregates. This data supports HSPGs and HS GAGs as human neural lineage differentiation and specification markers that may enable better isolation of human neural lineage-specific cell populations and improve our understanding of human neurogenesis.

## 1. Introduction

Adult neurogenesis is initiated by neural stem cells (NSCs) in selective neurogenic niches within the mammalian brain [1]. NSCs are the most primitive cell type in the central nervous system capable of self-renewal and responsible for the generation of mature neural cell types, including neurons, astrocytes and oligodendrocytes [2]. As NSCs undergo neurogenesis, they progress through numerous stages, including the transition into radial glial cells and neural progenitor cells (NPCs), and then mature into neuron, astrocyte or oligodendrocyte lineages. Throughout each stage of neurogenesis, distinct progenitor cell types can be identified by specific and often overlapping marker expression profiles [3,4,5]. While neurogenesis has been extensively studied to elucidate the mechanisms of neuronal differentiation, the development and characterisation of astrocyte models remain relatively underexplored. Given the growing recognition of dysfunctional astrocytes in the pathogenesis of neurodegenerative diseases [6], it is critical to establish robust and reproducible astrocyte cell models. However, significant challenges persist in understanding the molecular cues and environmental conditions that drive NPCs toward a glial fate. As a result, the efficient and reliable generation of functional astrocytes in vitro continues to be a complex and unresolved task. This is likely a result of our knowledge of human neurogenesis has been extrapolated from rodent studies [7], with species-specific differences hindering the translation of findings into human cells. Thus, a more comprehensive understanding of the regulatory mechanisms governing human NPCs, along with their marker profiles during astrocyte lineage specification and subsequent differentiation, is essential for understanding and recapitulating both normal and pathological glial functions.

Adult neurogenesis occurs in two discrete neurogenic niches within the brain, the subventricular zone (SVZ) and the subgranular zone (SGZ) [8]. Heparan sulfate proteoglycans (HSPGs), including the cell surface-bound syndecans (SDCs) and glypicans (GPCs), are ubiquitously expressed macromolecules and major constituents of the NSC niche [9]. Within the neurogenic niches, HSPGs interact with numerous signalling molecules via their highly diverse HS glycosaminoglycan (GAG) side chains [10]. GAGs possess structural complexity mainly due to their highly variable sulfation pattern, a result of a complex post-translational biosynthesis process [11]. Their GAG chains aid neural stem cell self-renewal, proliferation, lineage specification, differentiation and maintenance [12,13]. Work by our team and others have suggested HSPGs as biomarkers of human neural fate decisions to better identify, define and isolate neural subpopulations [14,15].

As essential regulators of neurogenesis, growth factors are often used in an auxiliary strategy to improve the properties of in vitro neural cultures [16]. The HS-binding proteins [17,18,19], brain-derived neurotrophic factor (BDNF) and platelet-derived growth factor (PDGF) are neuroregulatory growth factors often used in neural inductive cultures for the specific lineage differentiation of various stem cell types [20,21,22,23,24,25]. BDNF is a neurotrophin with a diverse range of effects during brain development, including the promotion of neural cell proliferation, differentiation, maturation, survival and synaptic plasticity [26,27,28,29,30]. Interestingly, studies have demonstrated that BDNF may support serotonergic neuron differentiation in mice via the induction of S100 calcium-binding protein B (S100B) secretion from astrocytes [31]. Astrocytes have been shown to express higher levels of BDNF when compared with neuronal lineages [32], although overall BDNF expression remains relatively low in all human neural cells [33]. While the majority of the functions of BDNF are associated with promoting neurogenesis—often mediated through astrocytic secretions—PDGF has been implicated in not only neurogenesis and dendritic development but also in gliogenesis, highlighting its broader role in neural lineage specification [34,35,36,37]. The interactions between BDNF and PDGF with HSPGs are not well understood in the context of human neurogenesis, with investigations into their interactions likely to provide improved methods for the identification and differentiation of NPCs.

We have previously shown that HSPGs are critical proteins in the maintenance and neural differentiation of embryonic stem cell (ESC)-derived human neural stem cells (hNSC H9 cells) [14,15], cortex-derived normal human neural progenitor cells (hnNPCs) [37] and human mesenchymal stem cell (hMSC) populations [38]. Specifically, GPC-1 and -4 were found to be novel markers of neuronal differentiation, as validated through the targeted small interfering RNA (siRNA) knockdown (KD) of GPC1 and GPC4 in the short-term (15–18 days) and long-term (40–60 days) neuronal differentiation of hNSC H9 cell populations [14,15]. GPC1 and GPC4 were also observed to be differentially expressed along the neurites and cell bodies of long-term differentiating neurons, with a higher GPC4:GPC1 expression ratio observed in response to BDNF treatment, and lowered GPC4 expression in PDGF-treated cultures [15]. In addition, the siRNA KD of GPC1 and GPC4 resulted in reduced neuronal marker expression and altered responsiveness to BDNF and PDGF [15], demonstrating the importance of these HSPGs in human neurogenesis.

This study aimed to identify human lineage-specific HSPG signatures that may serve as markers of glial cell differentiation, particularly in response to distinct neuroinductive conditions. The overarching goal was to establish a robust and reproducible method for generating functional astrocyte cultures in vitro, using commercially available human NPC lines. The use of cell lines provides the advantage of maintaining a stable phenotype across time in culture (passages) and overcomes the limited cell expansion capability associated with primary cell populations [39]. The human neural cell lines used included: the SH-SY5Y human neuroblastoma cell line, the human-induced NSC lines ReNcell VM (ventral mesencephalon-derived) and ReNcell CX (frontal cortex-derived). The SH-SY5Y human neuroblastoma cell line, commonly differentiated into mature neuronal cells, serves as a well-characterised, neuronally committed model resembling immature catecholaminergic neurons [40,41]. As such, it was included as a neuronal control to facilitate the comparison of marker expression and HSPG profile changes. The two ReNcell lines are stable and commercially available models, shown to exhibit distinct electrophysiological properties following differentiation. This is likely due to differences in their derivation methods—with ReNcell VM cells being derived from bulk foetal brain tissue, while ReNcell CX cells originate from a single cell clone [39,40,41]. The ReNcell CX cell line shares similar neural marker expression, neuronal and glial lineage potential and characteristics in vitro with hESC-derived hNSCs [42]. Microarray analysis has shown ReNcell CX cells to differentially express ~12.4% of all genes compared with hESC-hNSCs, and to be more distant from hESCs when compared with hNSCs using principal component analysis [42,43], suggesting that ReNcell CX cells possess lower stemness and more NPC-like characteristics. Although both lines are capable of differentiating into mature neural lineages, the resulting cultures tend to be heterogeneous, with neural subtypes emerging stochastically upon the withdrawal of growth factors such as fibroblast growth factor (FGF) and epidermal growth factor (EGF) [39]. While ReNcell CX cells offer a relatively uniform and controlled model of neural differentiation, the inherent heterogeneity of ReNcell VM cultures more closely reflects the cellular diversity present in the neurogenic niches of the developing and adult brain. As such, the ReNcell VM line serves as a valuable model for studying molecular dynamics, such as HSPG expression changes, in a physiologically relevant and complex environment. The use of both these ReNcell models, allows the exploration of the influence of lineage-specific HSPGs on progenitor maintenance and neural differentiation in mixed populations.

In this study, the cell lines were first cultivated under their specific optimal basal conditions to prevent the introduction of any bias from changes in culture conditions. We aimed to establish a defined and reproducible astrocyte induction protocol for the heterogenous ReNcell VM NPC line, with differentiated SH-SY5Y cells serving as a neuronal comparator and ReNcell CX-derived astrocytes as a glial benchmark. By profiling key HSPG markers, we investigated how BDNF and PDGF influence lineage specification and glial commitment. These HSPG signatures may not only serve as markers of lineage progression but also provide actionable targets for steering NPC fate—either through selective molecular interventions or the design of small molecules modulating HS side chain structures for tailored neural subtype generation.

## 2. Materials and Methods

### 2.1. Human Neural Cell Line Propagation and Neural Lineage Differentiation

#### 2.1.1. SH-SY5Y Human Neuroblastoma Cell Line

The SH-SY5Y cell line (RRID:CVCL_0019) was routinely cultured as a monolayer in Basal Growth Medium (DMEM/F-12 + GlutaMAX-I (Gibco, Waltham, MA, USA, Cat #: 10565018)) supplemented with 10% foetal bovine serum (FBS; Serana, Bunbury, Western Australia, Australia, Cat# S-FBS-AU-015) and 1% penicillin/streptomycin (pen/strep; Lonza, Basel, Switzerland, Cat# 17-602E)). The SH-SY5Y cells were differentiated toward the neuronal lineage using the protocol as described previously and from here on will be referred to as neuronal inductive cultures [44]. Briefly, cells were plated onto 35 mm culture dishes at 1.0 × 10^5^ cells/dish in SH-SH5Y Basal Growth Media and allowed to attach overnight. Warmed and pH-equilibrated (37 °C, 5% CO_2_) SH-SY5Y Differentiation Media #1, comprising DMEM/F-12 + GlutaMAX-I supplemented with 2.5% heat-inactivated FBS (hiFBS), 1% pen/strep and 10 μM all-trans retinoic acid (ATRA; Sigma-Aldrich, St. Louis, Missouri, United States Cat# R2625), was then added to replace the growth media, initiating Day 1 (D1) of differentiation. Media was changed using Differentiation Media #1 on D3 and D5. On D7, cells were replated onto fresh 35 mm dishes in Differentiation Media #1 and allowed to reattach overnight. Media was changed to warmed and equilibrated Differentiation Media #2 (DMEM/F-12 + GlutaMAX-I supplemented with 1% hiFBS, 1% pen/strep and 10 μM ATRA) on D8. On D10, cells were replated onto Geltrex^TM^ (Thermo Fisher Scientific, Waltham, MA, USA, Cat# A1413202)-coated 24-well plates in Differentiation Media #2. Warmed and equilibrated Differentiation Media #3 (Neurobasal media (Gibco^TM^, Cat# 21103-049) supplemented with 1X B-27^TM^ Supplement (Gibco^TM^, Cat# 17504044), 20 mM KCl, 1% pen/strep, 2 mM GlutaMAX-I^TM^ (Gibco^TM^, Cat# 35050-061), 50 ng/mL BDNF (Invitrogen^TM^, Carlsbad, CA, USA, Cat# PHC7074), 2 mM db-cAMP (STEMCELL^TM^ Technologies, Vancouver, British Columbia, Canada, Cat# 73884) and 10 μM ATRA) was used to replace the media on D11, then two further media changes were performed on D14 and D17. Homogenous, mature neuronal cultures were ready for more detailed examination on D18.

#### 2.1.2. ReNcell CX Human Neural Progenitor Cell Line

The ReNcell CX cell line (EMD Millipore, Billerica, MA, USA, Cat# SCC007, RRID:CVCL_E922) was routinely expanded as a monolayer on laminin-coated (20 μg/mL; Invitrogen^TM^, Carlsbad, CA, USA, Cat# 23017-015) culture vessels in DMEM/F-12 + GlutaMAX-I supplemented with 20 ng/mL FGF-2 (Gibco^TM^, Carlsbad, CA, USA, Cat# PHG0024) and 20 ng/mL epidermal growth factor (EGF; Gibco^TM^, Carlsbad, CA, USA, Cat# PHG0314). Differentiation towards the astrocyte lineage was performed by first withdrawing growth factors (FGF-2, EGF) and allowing spontaneous differentiation (SD) for 3 days on Geltrex^TM^-coated culture vessels. Differentiating ReNcell CX cells were then transferred to uncoated culture vessels (tissue culture plastic) in astrocyte differentiation media consisting of DMEM medium supplemented with 2% FBS, 2% N-2 Supplement (Gibco^TM^, Carlsbad, CA, USA, Cat# 17502048) and 2 mM GlutaMAX-I (media formulation from Thermo Fisher Scientific, Waltham, MA, USA, Cat# 35050061) for up to 2 weeks (14 days). Astrocyte differentiation media was changed every 3–4 days. ReNcell CX differentiation cultures from here on will be referred to as ReNcell CX astrocyte inductive cultures.

#### 2.1.3. ReNcell VM Human Neural Progenitor Cell Line

The ReNcell VM cell line (EMD Millipore, Billerica, MA, USA, Cat# SCC008, RRID:CVCL_E921) was maintained as a monolayer on laminin (20 μg/mL)-coated 10 cm^2^ plates in ReNcell NSC Maintenance Medium (EMD Millipore, Billerica, MA, USA, Cat# SCM005) supplemented with 20 ng/mL of FGF-2 and 20 ng/mL of EGF. For neural differentiation, ReNcell VM cultures underwent unguided/spontaneous differentiation on Geltrex-coated plates initiated by withdrawal of growth factor (FGF-2, EGF), which resulted in an untreated (UT) heterogeneous neural culture examined at D14. These cultures will be referred to as UT/spontaneous ReNcell VM neuroinductive cultures. Additionally, ReNcell VM neural differentiation cultures were supplemented with neuroregulatory growth factors, 10 ng/mL of BDNF and 10 ng/mL platelet-derived growth factor BB (PDGF-BB; Sigma-Aldrich^®^, St. Louis, MI, USA, Cat# P4056-50UG) to direct lineage specification. The following culture conditions were examined: UT/spontaneous differentiation (spontaneous ReNcell VM neuroinductive cultures), +BDNF and +PDGF. Growth factors were re-applied to the cells every 3–4 days during media changes. The generated neural cultures were examined on D14.

### 2.2. Viability Assay

The cell viability of cell line-derived neural specific lineage cultures was assessed by fluorescein diacetate (FDA; live)/propidium iodide (PI; dead) staining. FDA (Sigma-Aldrich^®^, St. Louis, MI, USA, Cat# F7378) and PI (Sigma-Aldrich^®^, St. Louis, MI, USA, Cat# 81845) were added to unfixed neural cultures at 1:1000 dilution and incubated at room temperature for 10 min in the dark. Detection was performed on an Olympus IX81 (Olympus Corporation, Tokyo, Japan) phase-contrast fluorescent microscope using FITC (FDA/live) (excitation = 490 nm, emission = 530 nm) and Texas Red (PI/dead) (excitation = 590 nm, emission = 620 nm) filters. Live/dead percentages were generated by using Volocity Cellular Imaging and Analysis Software v6.3 (Quorum Technologies Inc., Sacramento, CA, USA) to count FDA-positive cells and PI-positive nuclei.

### 2.3. Fluo-4 Calcium Signalling Assay

Neuronal calcium signalling, indicative of the critical function of neurotransmission, was examined using the fluorescent calcium indicator, Fluo-4 Direct^TM^ (Invitrogen^TM^, Carlsbad, CA, USA, Cat# F10471), as described previously [15]. Briefly, all cells tested for calcium signalling were plated at the appropriate seeding density in 96-well (0.32 cm^2^) plates. Next, 2× Fluo-4 Direct^TM^ calcium reagent loading solution was prepared by adding Hanks’ Balanced Salt Solution (HBSS) with 2 mM CaCl_2_, 0.1 mM MgCl_2_ and 250 mM probenecid. Cells cultured on 96-well plates were incubated with an equal volume of culture media of Fluo-4 (1× final concentration) for 45 min at 37 °C then allowed to cool to room temperature (approx. 20 min) prior to imaging. Fluo-4 Direct^TM^ has an Em λ = 494 nm and Ex λ = 516 nm and was detected using the fluorescein isothiocyanate (FITC) filter on an Olympus IX81 microscope with a Hamamatsu Orca Camera using the Volocity Image Capture and Analysis Software v 6.3. Cultures were imaged for 200 s at 0.8 s intervals, equating to 251 captured time-points. At each time-point, the fluorescence level of ten (10) regions of interest (ROIs) were measured using Volocity. Fluorescence signal intensity (FI) was normalised to time-point 1 (F0) and FI/F0 in Microsoft Excel and was graphed using the ‘stackedplot’ function in MATLAB (R2018b, RRID:SCR_001622).

### 2.4. Total RNA Isolation, cDNA Synthesis and Q-PCR

Gene expression analysis was carried out as previously described [45]. Briefly, RNA isolation was carried out using a Direct-zol™ RNA MiniPrep kit (Zymo Research Corp., Irvine, CA, USA, Cat# R2052) with in-column DNase I (300 U/rxn) treatment. RNA conversion to cDNA was performed using the iScript cDNA Synthesis Kit (Bio-Rad, Hercules, CA, USA, Cat# 170-8891) following the manufacturer’s instructions. Briefly, 150 ng of RNA template was reverse transcribed using 4 μL of 5× iScript Reaction Mix, 1 μL of iScript Reverse Transcriptase and nuclease-free water to a total reaction volume of 20 μL. Samples were incubated for 5 min at 25 °C (priming), 20 min at 46 °C (reverse transcription), 1 min at 95 °C (Reverse Transcriptase inactivation) and then held at 4 °C.

Human neural cultures generated in this study were characterised for the gene expression of the HSPG core protein, GAG biosynthesis machinery, as well as pluripotency and neural-specific lineage markers. Q-PCR was performed in 384-well plates using the Applied Biosystems^®^ QuantStudio-7 Flex Real-Time PCR System (Life Technologies, Carlsbad, California, United States) where three biological replicates per culture condition was assayed in technical quadruplicate. For each reaction, 120 ng of cDNA was amplified with 5 μL of 1× SYBR Green Mix (Promega Corporation, Madison, Wisconsin, United States, Cat# A6002), 200 nM of each primer pair and 0.1 μL ROX Passive Reference Dye (Promega) in a final reaction volume of 10 μL. The cycling conditions were as follows: 50 °C × 2 min, 95 °C × 3 min, followed by 50 cycles at 95 °C × 3 s, 60 °C × 30 s. Gene expression was normalised against the expression of rRNA 18*S*, an endogenous control gene. All primers used in this study have been validated in several previous studies from the research group [14,15,38,45,46,47]. The primer sequences are available in Appendix A.

Gene expression data from neural cultures were combined between experimental replicates and analysed using the 2^−ΔCt^ method. Fold changes in target gene expression were calculated between basal and differentiated cultures. Growth factor-modulated cultures were analysed as fold changes compared with untreated (UT) differentiated cultures. Data is presented as Log2FoldChange vs. Log(p-value) on volcano plots. Student’s *t*-test was used to determine significant changes in target gene expression levels between basal and differentiated cultures, and between culture conditions. Significance was set at α = 0.05. Volcano plots were drawn using GraphPad Prism 9 software (Version 9.1.0, RRID:SCR_002798).

### 2.5. Immunocytochemistry (ICC) Analysis

To fix cells for immunostaining, cells on coverslips (13 mm, plastic) in 24-well plates (Sarstedt AG & Co. KG, Nümbrecht, Germany, Cat# 83.3922) were first partially fixed by gently removing 70% of the media, which was then replaced with 4% paraformaldehyde (PFA) for 2 min, followed by a complete fix with fresh 4% PFA for 20 min at room temperature (RT). Cells were permeabilised and antigen blocking was performed through incubation with 2% normal donkey serum (NDS) + 0.3% Triton-X in PBS with Ca^2+^ and Mg^2+^ for 3 h at RT with gentle rotation. Primary antibodies diluted in 2% NDS in PBS were added to the cells and incubated, static, overnight at 4 °C. The primary antibodies and dilutions used were anti-SOX2 (1:1000; Millipore, Billerica, MA, USA, Cat# AB5603, RRID:AB_2286686), anti-MAP2 (1:200; Abcam, Cambridge, UK, Cat# ab36447, RRID:AB_776175), anti-GFAP (1:250; Abcam, Cambridge, UK, Cat# ab7260, RRID:AB_305808), anti-O1 (1:500; Millipore, Billerica, MA, USA, Cat# MAB344, RRID:AB_94860) and anti-HS 10E4 (1:1000; Amsbio, Cat# 370255-1, RRID:AB_10891554). The next day, primary antibodies were removed, cells were washed for 3 × 10 min with washing solution (0.2% NDS + 0.03% Triton-X in PBS) and secondary antibodies diluted in 2% NDS in PBS were then applied to the cells for 2 h at RT. Following the 3 × 10 min washes with washing solution (0.2% NDS + 0.03% Triton-X in PBS), cells on coverslips were placed on glass microscope slides with cells facing up and mounted using ProLong^TM^ Gold AntiFade Mountant with 4′,6-diamidino-2-phenylindole (DAPI) counterstain (Life Technologies, Cat# P36941) with another coverslip placed on top of the cells. Slides were imaged using an Olympus IX81 inverted phase-contrast fluorescence microscope with a Hamamatsu Orca R2 Camera using the Volocity v6.3 Image Capture and Analysis Software (Quorum Technologies Inc., RRID:SCR_002668) at 20× magnification. The image contrast, brightness and sharpness of images acquired were processed using Fiji (ImageJ2 v2.1.0/1.53c, RRID:SCR_002285) software. Quantitation of the percentage of signal-positive cells was performed by manual cell counting, where three individual images from each experimental replicate (*n* = 3) were used for quantitation. The results are presented as mean values with standard deviations (SD) with statistical testing performed using unpaired *t* tests with Welch’s correction. For differentiated ReNcell VM neural cultures, due to the abundance of neuronal protrusion and cell numbers, ICC images were quantitated using signal intensity normalised to cell numbers.

### 2.6. Western Blotting

Western blot (WB) analysis was conducted using biological triplicates to assess HSPG core protein expression during ReNcell VM differentiation in the presence of BDNF and PDGF. For detection of the 3G10 neo-epitope, lysates from all triplicates were pooled to ensure sufficient protein yield, particularly for the experiments presented in Figure 6. In all other experiments, GPC2- and SDC3-specific antibodies were used to detect core protein expression in independent biological triplicates without pooling. The 3G10 neo-epitope is only generated following the digestion of HS GAGs by heparitinase, where the 3G10 antibody reacts with desaturated hexuronate “stubs” that remain on the core proteins [48]. Thus, HSPG core protein forms can be visualised by 3G10 staining, detected as multiple bands on WBs. The sizes of the HSPG cell surface core proteins are as follows (from GeneCards^®^ The Human Gene Database, Weizmann Institute of Science, Rehovot, Israel): syndecan-1 (SDC1; 32 kDa), syndecan-2 (SDC2; 22 kDa), syndecan-3 (SDC3; 45 kDa), syndecan-4 (SDC4; 22 kDa), glypican-1 (GPC1; 62 kDa), glypican-2 (GPC2; 63 kDa), glypican-3 (GPC3; 66 kDa), glypican-4 (GPC4; 62 kDa), glypican-5 (GPC5; 64 kDa) and glypican-6 (GPC6; 63 kDa). While the reliance on molecular weight alone to identify specific core proteins recognised via HS stub detection is a limitation of this approach, the antibody provides a putative range of expected bands consistent with various HSPGs. However, the confirmation of individual core protein identities would require validation using protein-specific antibodies. WB analysis of human neural cell line basal and neural inductive cultures was conducted as previously described [15,37,38,39,40,41,42,43,44,45,49]. Briefly, technical triplicates of cells grown in basal and neural inductive culture conditions of each cell line were pooled prior to protein collection and total protein extracted using RUNX protein lysis buffer with a protease and phosphatase inhibitor (1:100; Cell Signaling Technology^®^, Danvers, MA, USA, Cat# 5872). Protein samples were quantitated using the Qubit Protein Assay Kit (Thermo Fisher Scientific, Waltham, MA, USA, Cat# Q33211) and a Qubit 2.0 Fluorometer (Thermo Fisher Scientific, Waltham, Massachusetts, United States). Approximately 20–30 μg of protein was separated by SDS-PAGE using 12% TGX^TM^ FastCast^TM^ Acrylamide gels (Bio-Rad, Hercules, CA, USA, Cat# 1610175) and transferred onto 0.2 μm PVDF membranes (EMD Millipore Corp., Billerica, MA, USA, Cat# ISEQ00010) using the Transblot turbo system (Bio-Rad). The membrane was blocked with 5% skim milk in TBST (Tris-buffered saline + 0.1% Tween-20) followed by the addition of primary antibodies diluted in 5% bovine serum albumin (BSA) in TBST and incubated overnight at 4 °C. The primary antibodies used include: anti-Δ3G10 (1:1000; US Biological, Salem, MA, USA, Cat# H1890-75, RRID:AB_2722745), anti-GPC2 (1:2000, Invitrogen, Carlsbad, CA, USA, Cat#PA5-115299, RRID:AB_2899935) and anti-SDC3 (1:1000; abcam, Cambridge, UK, Cat# ab63932, RRID:AB_1143216) with GAPDH (1:1000; Cell Signaling, Danvers, MA, USA, Cat# 2118, RRID:AB_561053) used as the loading control. Primary antibodies were removed the next day, membranes were washed in TBST, followed by incubation with HRP/fluorophore-conjugated secondary antibodies (1:3000; anti-Mouse IgG, Cell Signaling, Danvers, MA, USA, Cat# 7076, RRID:AB_330924; anti-Rabbit IgG, Cell Signaling, Danvers, MA, USA, Cat# 7074, RRID:AB_2099233; 1:10,000; Alexa Fluorophore 488 anti-Mouse IgG, Thermo Fisher Scientific, Waltham, MA, USA, Cat#A10680, RRID:AB_2534062 and 1:10,000; Cyanin 3 anti-Rabbit IgG, Thermo Fisher Scientific, Waltham, MA, USA, Cat#A10520, RRID:AB_10563288) for 2 h at room temperature. Target protein detection was performed with enhanced chemiluminescence (Clarity ECL, Bio-Rad, Hercules, CA, USA, Cat# 1705060) using the Fusion FX Spectra chemiluminescence system (Vilber Lourmat, Fisher Biotec, Wembley, Perth, Western Australia, Australia) and each blot quantitated using Image J software (version 1.52q, NIH, RRID:SCR_003070). Where appropriate, proteins with non-overlapping molecular weights were probed on the same blot.

## 3. Results

### 3.1. Human Neural Cell Line Cultures Adapt Neural Cell Morphology, Maintain High Cell Viability and Exhibit Spontaneous Calcium Activity Following Neuroinductive Cultures

The three neural cell lines studied, SH-SY5Y, ReNcell CX and ReNcell VM, were grown as monolayers under basal expansion conditions as described above. Basal SH-SY5Y cells grew in clusters with non-polarised cell bodies and few condensed processes (Figure 1a). During SH-SY5Y neuronal induction, cell bodies became pyramidal with neurite-like extensions. Adherent spheres were observed at the endpoint of this study (D18) with inter-spherical connections observed (Figure 1b). ReNcell CX cells were polygonal in morphology during basal culture with several protrusions observed; at high density these cells had a cobblestone pattern (Figure 1d). When examined following astrocyte inductive culture, astrocyte-like processes were observed within the cultures, with cells displaying a “star-like” morphology by Day 14 (D14) of differentiation (Figure 1e). There was a lack of secondary and tertiary structures such as branches and peripheral fine processes observed in these cultures, characteristics of mature astrocytes [49]. Undifferentiated ReNcell VM cells changed from a “paving stone” morphology at the start of the neuroinductive culture period (Figure 1g) to a classic neuron-like morphology with small cell bodies and visibly elongated dendritic-like processes after D14 of neuroinductive culture (Figure 1h). All three human neural cell lines maintained high viability as analysed by FDA/PI staining, with SH-SY5Y neuronal cultures maintaining an average of 98% viability, the ReNcell CX astrocyte inductive cultures maintaining an average of 97% viability and the ReNcell VM spontaneous neuroinductive cultures maintaining an average of 94% viability (Figure 1c,f,i,j).

Fluo-4 calcium indicator dye was added to cell line-derived neural lineage inductive cultures to detect any spontaneous calcium oscillation, a characteristic of neural cell cultures and an indirect method of examining spontaneous action potentials representative of functionality [50]. Calcium signalling activity in cell line-derived neural specific cultures was compared to the hNSC H9 neuronal cultures at D60, with known oscillatory activity [33]. In the positive control cultures, hNSC H9 D60 neuronal cells exhibited a “wave-like” pattern of calcium signals over time, indicative of spontaneous calcium oscillation (Appendix A, Online Resource 1). In a non-neural cell type, such as the human breast cancer (HBC) cell line MCF-7, the slow uptake of Fluo-4 was observed, indicating a lack of cell-cell communication via calcium signalling (Appendix A, Online Resource 2). The cell line-derived neural lineage-specific cultures, the SH-SY5Y neuronal inducted cultures at D18, displayed a similar wave pattern of calcium signal to the hNSC H9 neuronal D60 cultures (Figure 1k, Online Resource 3), indicating spontaneous calcium oscillations. Calcium signals from ReNcell CX astrocyte inductive D14 cultures were observed to be similar to those reported for astrocyte cultures, described as global waves, local waves and microdomains [51], which together appear as calcium bursts and intrinsic fluctuations [52] (Figure 1l ROIs 5 and 7; Online Resource 4). Interestingly, in the ReNcell VM spontaneous neuroinductive cultures, the slow uptake of Fluo-4 was observed, similar to the uptake observed in MCF-7 HBC-negative control cultures, suggesting a lack of neural functionality, likely due to a mixture of non-committed NPC populations within the cultures (Figure 1m, Online Resource 5).

### 3.2. SH-SY5Y Neuronal Differentiated Cultures Exhibit Similar Characteristics to Differentiated hNSC H9 Cells

The SH-SY5Y neuronal inductive cultures were then further characterised for neural lineage marker expression and changes in HSPG profile. The ICC analysis of SOX2, a key transcription factor in NSC maintenance [53], was shown to be widely expressed by basal SH-SY5Y cells (Figure 2a), and to be reduced and lowly expressed in the neuronal inductive cultures (Figure 2b). SOX2 localisation can be nuclear and/or cytoplasmic depending on cell type, self-renewal (stem cell) or lineage (differentiation) state [54]. The cytoplasmic localisation of SOX2 was observed in SH-SY5Y cells, likely due to the acetylation of SOX2, previously reported in ESC differentiation [55] and observed in a human mesencephalic cell line [56]. The cytoplasmic SOX2 expression observed in the basal SH-SY5Y cultures is reminiscent of NPCs. The quantitation of SOX2 expression showed a significant decrease (*p* = 1.28 × 10^−5^) in the number of SOX2+ cells following SH-SY5Y neuronal induction (86 ± 24% reduced to 2 ± 5%) of SOX2+ cells (Figure 2c). The expression of MAP2, the mature neuronal marker, was observed to be significantly increased (*p* = 1.69 × 10^−5^) within the SH-SY5Y neuronal cultures, with 0.8 ± 0.9% MAP2+ cells to 80 ± 12% MAP2+ cells detected by D21 (Figure 2d–f). Interestingly, SH-SY5Y basal cultures expressed high levels of glial lineage markers, which were observed to be maintained throughout the neuronal inductive culture (Figure 2g,h). The expression of the astrocyte marker GFAP was found to be maintained at 94 ± 9% and 98 ± 3% GFAP+ cells in basal and inductive cultures, respectively (Figure 2i). O1 is considered a marker of late oligodendrocyte progenitors [57], with undifferentiated SH-SY5Y cells stained strongly for this marker (Figure 2j; 87 ± 22% O1+ cells), with staining found to be maintained at a lower level (58 ± 24% O1+ cells) in the inductive cultures (Figure 2k,l).

The gene expression analysis showed a significant downregulation of several stem cell and neural lineage markers, including the pluripotency markers (*POU5F1*; *p* = 0.0016, and *NANOG*; *p* = 8.06 × 10^−5^), NPC markers (*SOX1*; *p* = 0.0011, *SOX2*; *p* = 2.69 × 10^−5^, and *NES*; *p* = 0.0154), astrocyte markers (*S100B*; *p* = 0.030, *SLC1A3*; *p* = 0.0115) and oligodendrocyte marker (*GALC*; *p* = 0.0457) (Figure 2m). Interestingly, the gene expression level of the neuronal markers, *NEFM* (*p* = 0.0003) and *MAP2* (*p* = 0.0027), was found to be significantly downregulated (Figure 2m). In contrast, the neuronal marker *TUBB3* (*p* = 0.0297) was observed to be significantly upregulated, along with the oligodendrocyte marker *OLIG2* (*p* = 0.0475; Figure 2m). Overall, the SH-SY5Y neuronal inductive cultures displayed a neuronal phenotype, evident from the cellular morphology, calcium activity and the gene/protein expression of neuronal markers.

### 3.3. Glypican 2 Predominant in Neuronal Differentiated Cultures

Amongst the HS GAG biosynthesis and modification enzymes, the HS chain elongation enzyme *EXT2* (*p* = 0.0059) as well as 6-*O*-endo-sulfatase *SULF2* (*p* = 8.32 × 10^−9^) were found to be significantly upregulated in neuronal inductive SH-SY5Y cultures, indicating an increased length of HS GAGs and/or the number of HS chains within these cultures, as well as modifications at the 6-*O*-sulfation sites. Furthermore, the gene expression levels of several *N*-deacetylase and *N*-sulfotransferases, *NDST1* (*p* = 3.39 × 10^−6^), *NDST2* (*p* = 5.78 × 10^−5^) and *NDST4* (*p* = 0.0036), were observed to be significantly downregulated, along with the HS 6-*O*-sulfotransferase, *HS6ST1* (*p* = 0.0006) (Figure 2n) in neuronal inductive SH-SY5Y cultures. This data suggested a decrease in sulfation requirement at the *N*- and 6-*O*-sulfation sites following SH-SY5Y neuronal induction with the medication of *N*- and 6-*O*-sulfation sites reported during stem cell neural differentiation [58]. The ICC analysis of HS GAG content of SH-SY5Y cultures was performed using the pan-HS 10E4 antibody. The 10E4 epitope contains *N*-sulfated glucosamine residue/s which react with the antibody allowing the detection of all HS GAG chains [59,60]. In basal SH-SY5Y cells, 10E4 was demonstrated to have punctuated expression and as cells differentiated towards the neuronal lineage, 10E4 expression became more widespread and localised to the cell surface (Figure 2s). The quantitation of 10E4 signal intensity showed a >2-fold increase (*p* = 0.0098) in the intensity of 10E4 staining in neuronal inductive cultures when compared with basal SH-SY5Y cultures (Figure 2r) demonstrating the acquisition of neuronal phenotypes coinciding with increased GPC expression and HS GAG content.

There was a significant upregulation of *GPC2* (*p* = 0.04996) gene expression, along with the significant downregulation of *SDC1* (*p* = 0.0149) (Figure 2n) in the neuronal inductive SH-SY5Y cultures. The WB analysis of polyclonal GPC2 and SDC3 in SH-SY5Y-derived neuronal cultures revealed an interesting core membrane-bound HSPG protein profile post lineage induction. In basal SH-SY5Y cells, all members of the SDC3 family, including membrane-bound SDC3 (MB SDC3, 45 kDa), truncated membrane-bound SDC3 (TMB SDC3, 19 kDa) [61] and cytoplasmic domain SDC3 (CD SDC3, 12 kDa) [62], were found to be expressed. However, in the inductive neuronal SH-SY5Y cultures, glycosylated or neural GPC2 (100 kDa) was observed to be the predominant form, along with membrane-bound GPC2 (MB GPC2, 63kDa) (Figure 2o). The quantification of WB signals indicated the reduced expression of all SDC3 family members (MB, TMB, CD) in the neuronal cultures (Figure 2p), correlating with the observed significant decrease in SDC3 gene expression. Notably, SDC3 exhibited the lowest relative gene expression levels in neuronal cultures when compared with the basal cultures (Figure 2n). Interestingly, while MB GPC2 expression was found to be lower in the neuronal inductive cultures, neural GPC2 was also found to be highly expressed in these cultures, but undetected in SH-SY5Y basal cultures. As a result, total GPC2 expression in neuronal cultures exceeded that of the basal cultures (Figure 2q), aligning with the observed significant upregulation of GPC2 gene expression (Appendix A).

### 3.4. ReNcell CX Astrocyte Induction Conditions Produce an Immature Astrocyte Culture in 14 Days

The ICC analysis showed the NSC marker SOX2 to be widely expressed by ReNcell CX basal cells (Figure 3a), comparable to SH-SY5Y basal cultures, and lower in ReNcell CX astrocyte inductive cultures (Figure 3b). The quantitation of this data identified 85 ± 9% SOX2+ cells in basal cultures, with a significant reduction (*p* = 0.0012; 49 ± 16%) in SOX2+ cells in the astrocyte inductive cultures (Figure 3c). The neuronal marker MAP2 was not found to be expressed in basal ReNcell CX cultures, although a subpopulation (31% ± 20%, *p* = 0.0268) of MAP2+ cells were identified in the ReNcell CX astrocyte inductive culture conditions (Figure 3d,e). In addition, a significant increase in the expression of the astrocyte marker GFAP (*p* = 0.0043) was observed in ReNcell CX astrocyte inductive cultures (from 73 ± 16% to 96 ± 4% GFAP+ cells; Figure 3g–i). An examination of the oligodendrocyte marker O1 revealed that the basal ReNcell CX cultures contained 20 ± 19% O1+ cells, with 32 ± 15% O1+ cells observed in the astrocyte inductive culture of these cells (Figure 3j–l).

When compared with the basal cultures, the ReNcell CX astrocyte inductive cultures were found to express a significantly lower level of pluripotency and NSC gene markers, *POUF51* (*p* = 0.0195), *SOX2* (*p* = 0.0456) and *NES* (*p* = 0.0054), correlating with a decrease in SOX2+ cells (Figure 3m). The neuronal marker *NEFM* (*p* = 0.0144) was also found to be significantly downregulated (Figure 3m), confirming a global reduction in neuronal lineage potential within the astrocyte inductive culture conditions. During human brain development, the astrocyte gene *SLC1A3*, which encodes for excitatory amino acid transporter 1 (EAAT1), is primarily expressed by developing astrocytes with *SLC1A2* (EAAT2) expressed by adult astrocytes [63]. In the ReNcell CX astrocyte inductive cultures, SLC*1A3* (*p* = 0.0018) was found to be significantly downregulated, with the mature astrocyte marker, *SLC1A2*, expressed at a similar level to the basal cultures (Figure 3m). This suggested that cells within these inductive cultures have committed to the astrocyte lineage, yet remain immature; a longer time in culture is likely needed to produce a more mature, homogenous population of astrocytes. Indeed, several existing astrocyte differentiation protocols require at least 30 days or up to 4 weeks in culture to generate mature astrocytes [64,65,66,67].

### 3.5. Increase in HS Production Is Characteristic of Astrocytic Lineage

The significant downregulation of *NDST2* (*p* = 0.0143), *NDST4* (*p* = 0.0411) and *HS2ST1* (*p* = 0.0387) (Figure 3n) suggested a decrease or minimal change in both *N*- and 6-*O*-sulfation patterns following astrocyte lineage induction in the ReNcell CX cells. The HS elongation enzymes, *EXT1* (*p* = 0.0248) and *EXT2* (*p* = 0.0018), and the epimerisation enzyme, *C5-EP* (*p* = 0.0007), were found to be significantly upregulated in the astrocyte inductive cultures (Figure 3n), indicating increased GAG formation. The basal ReNcell CX cultures stained with the 10E4 antibody showed HS GAGs to be diffusely distributed across the cell surface, and as cells committed to the astrocyte lineage, these HS GAGs formed localised dense clusters along the cell surface (Figure 3r). This was supported by the quantitation of 10E4 signal intensity, which demonstrated that astrocyte inductive cultures have a significantly higher (*p* = 0.0018; 2.7-fold) HS GAG content when compared with the basal ReNcell CX cultures (Figure 3s). Following astrocyte lineage induction, the ReNcell CX cultures differentially expressed several HSPG core proteins, including the significant downregulation of *SDC1* (*p* = 0.0013), *SDC2* (*p* = 0.0013) and *GPC1* (*p* = 0.0005), and the significant upregulation of *SDC4* (*p* = 0.0018) and *GPC6* (*p* = 0.0417) gene expression (Figure 3n).

The WB analysis revealed distinct subtype-specific expression patterns for both GPC2 and SDC3 proteins in ReNcell CX cultures under basal and astrocyte inductive conditions. GPC2, a neural-specific isoform expression commonly enriched in neuronal precursors, was absent in the basal cultures but became weakly detectable under astrocyte inductive conditions, indicating a modest upregulation upon lineage specification. Interestingly, the MB isoform of GPC2 was found to be weakly expressed in both the basal and the astrocyte-induced cultures, although the signal remained faint and below the levels typically observed in highly neurogenic environments. These observations suggest that while astrocyte-inductive cues may transiently activate GPC2 expression, particularly of the neural isoform, this activation is relatively subdued, possibly reflecting an intermediate or transitional state rather than full neurogenic commitment. The low-level MB GPC2 expression in both conditions may reflect a basal role for GPC2 in maintaining minimal membrane signalling functions in progenitor cells, independent of lineage trajectory. This is supported by prior studies showing GPC2 interaction with Wnt and Hedgehog signalling components via its HS side chains, with its function tightly downregulated as progenitors transition to a glial lineage [68,69]. Moreover, the low neural GPC2 signal in the astrocyte inductive cultures may reflect the residual heterogeneity or limited co-expression of early neuronal markers during the initial phases of glial lineage priming. In contrast, SDC3 exhibited a divergent regulatory profile. While transcript levels remained largely unchanged between conditions, the WB analysis showed that both the MB and CD isoforms of SDC3 were undetectable in astrocyte inductive cultures. This suppression was specific to astrocyte-directed differentiation, as the basal conditions retained the clear protein expression of both isoforms. The loss of detectable SDC3 protein in the astrocyte cultures underscores its post-transcriptional regulation and aligns with its known role in neural progenitor maintenance, axonal pathfinding and response to growth factors such as FGF2 and EGF [70,71]. The coordinated downregulation of both MB and CD SDC3 also suggests a tightly controlled shut-off mechanism during glial commitment, potentially to enable the shift from progenitor plasticity to glial maturation. Together, these findings highlight the selective and isoform-specific modulation of GPC2 and SDC3 during astrocyte specification. While GPC2 showed a modest and possibly transitional upregulation of its neural isoform, SDC3 was robustly suppressed at the protein level, reflecting distinct regulatory dynamics underpinning neuroglial lineage segregation in human neural progenitor models.

### 3.6. ReNcell VM Mixed Neural Cultures Favour Neuronal Lineage, but Maintain NPC Subpopulation

The analysis of NSC (SOX2), neuronal (MAP2), astrocyte (GFAP) and oligodendrocyte (O1) markers by ICC indicated a reduced NSC phenotype in the ReNcell VM spontaneous neuroinductive cultures, accompanied by increased neural-specific lineage markers. A significant decrease (*p* = 8.73 × 10^−7^) in SOX2+ cells was observed between the basal (99 ± 1%) and the spontaneous neuroinductive (35 ± 12%) ReNcell VM cultures (Figure 4a–c), indicating that approximately two-thirds of the cells were lineage committed or differentiated, while a subpopulation remained as NPCs. Basal ReNcell VM cells do not express MAP2; however, following spontaneous neuroinduction, a small subset of MAP2+ cells (0.5%) was detected (Figure 4d–f), while GFAP and O1 signal intensity remained similar between the basal and spontaneous neuroinductive ReNcell VM cultures (Figure 4g–l).

The significant downregulation of NSC genes, *MSI1* (*p* = 0.0003) and *ENO2* (*p* = 1.83 × 10^−8^) further supported the reduction in NSC phenotype in the spontaneous neuroinductive cultures. However, *SOX2* (*p* = 0.0018) gene expression, along with the other NSC markers, *SOX1* (*p* = 0.0251) and *NES* (*p* = 0.0109), was observed to be significantly upregulated (Figure 4m). Corresponding with ICC data, neuronal marker *MAP2* (*p* = 0.0008), along with *NEFM* (*p* = 0.0427) and *DCX* (*p* = 0.0367) were found to be significantly upregulated at the gene expression level. While the ICC analysis indicated that the astrocyte and oligodendrocyte lineage markers maintained similar expression profiles in both basal and neuroinductive ReNcell VM cultures, the astrocyte markers (*GFAP*; *p* = 0.0048, *S100B*; *p* = 0.0012 and *SLC6A4*; *p* = 0.0292) along with the oligodendrocyte markers (*OLIG1*; *p* = 0.0288 and *OLIG2*; *p* = 0.0317) demonstrated significantly increased gene expression levels (Figure 4m). These observations suggest a heterogeneous mixture of different neural cell populations within the spontaneous ReNcell VM neuroinductive cultures. This is similar to findings reported by other groups, where spontaneously differentiated ReNcell VM cultures were demonstrated to comprise cells at varying stages of maturation and to express lineage fate markers of both neuronal and glial lineages [39,72].

### 3.7. Upregulation of Syndecan-1 Is Unique to ReNcell VM Mixed Neural Cultures

Heterogeneous 10E4 staining was observed in the basal ReNcell VM cultures to be localised to specific areas within the culture (Figure 4r). Following spontaneous neuroinductive culture, 10E4 staining showed a homogenous pattern throughout the culture, with expression detected on the cell surface and in cell clusters, with no increase in overall 10E4 signal intensity (Figure 4r,s) observed, suggesting that the distribution and localisation changed, but not the overall level of HS-GAGs within the cultures. The differences in the localisation of HS GAGs between the basal and inductive ReNcell VM neural cultures, in conjunction with the upregulation of HS enzymes, indicate active HS biosynthetic machinery throughout neural specification and the remodelling of the HS profile during lineage induction and differentiation.

Interestingly, the widespread significant upregulation of HSPG-related genes was observed in spontaneous neuroinductive cultures when compared with the basal ReNcell VM cultures. All surface-bound HSPG core proteins (*SDC1*; *p* = 0.0237, *SDC2*; *p* = 0.0025, *SDC3*; *p* = 0.0081, *SDC4*; *p* = 0.0299, *GPC1*; *p* = 0.0182, *GPC2*; *p* = 0.0283, *GPC3*; *p* = 0.0324, *GPC4*; *p* = 0.0005, *GPC5*; *p* = 0.0005 and *GPC6*; *p* = 0.0016) were found to be significantly upregulated following spontaneous neuroinduction, along with the majority of the HS biosynthesis and modification enzymes, except for *EXT2*, *SULF2* and *HPSE* (Figure 4n). The WB analysis to assess changes in GPC2 and SDC3 expression following ReNcell VM neuroinduction identified neural GPC2 and CD SDC3 expressed at very low levels in the basal ReNcell VM cultures. In contrast, MB GPC2 and all other SDC3 isoforms, including MB and TMB SDC3, exhibited higher expression levels in the basal cultures when compared with the spontaneously neuroinducted ReNcell VM cultures (Figure 4o–q). The upregulation of neural GPC2 and CD SDC3 was observed in these mixed neural cultures, resembling the expression patterns observed in the SH-SY5Y neuronal and ReNcell CX astrocyte inductive cultures. Notably, neural GPC2 expression was observed to be highest in the SH-SY5Y neuronal cultures, followed by the spontaneous neuroinductive ReNcell VM, and lowest in the ReNcell CX astrocyte cultures. These findings suggest that the highly glycosylated neural GPC2 correlated with increased neuronal marker expression.

### 3.8. HS-Binding Growth Factor BDNF Increased Neural Cell Numbers and PDGF Increased Oligodendrocyte Specficity

The ReNcell VM cells were then subjected to guided lineage specification in the inductive cultures via the supplementation of the HS-binding growth factors BDNF and PDGF-BB at 10 ng/mL. As previously described, our group has conducted extensive work with hNSCs, developing differentiation protocols via BDNF and PDGF-BB supplementation at 10 ng/mL [15]. The culture viability analysis by FDA/PI showed that the growth factor-supplemented cultures maintained high viability, as well as the BDNF-treated cultures (89%) and PDGF-treated cultures (95%; Figure 5a,b). The cell numbers at D14 of the ReNcell VM neural cultures with growth factor supplementation demonstrated that the BDNF-treated cultures contained the highest averaged cell numbers, approximately 3-fold higher than untreated cultures (Figure 5c).

The Fluo-4 calcium signalling analysis of ReNcell VM neural cultures supplemented with BDNF and PDGF exhibited minor calcium oscillations when examined at the D14 time-point (Figure 5d). Heterogeneity in the calcium signal was observed for both BDNF- (Online Resource 6) and PDGF-treated cultures (Online Resource 7). Several localised ROIs exhibiting “wave-like” calcium peaks were observed (ROIs 1 and 8 in BDNF-treated cultures, ROIs 7 and 9 in PDGF-treated cultures; Figure 5d), with the majority of cells within these cultures observed to only take up the Fluo-4 dye. However, in PDGF- treated cultures, higher calcium peaks (i.e., Figure 5d, ROI 7) were observed.

The neural phenotype of neuroinductive ReNcell VM cultures supplemented with BDNF and PDGF were further characterised by ICC and gene expression analyses. The percentage of SOX2+ cells in the BDNF- and PDGF-supplemented cultures (30–40%) were not observed to be significantly different from UT or spontaneous ReNcell VM neuroinductive cultures (Figure 5e,f), suggesting that growth factor modulation did not alter the proportion of progenitor cells. The neuronal marker MAP2 was found to be expressed in a minor percentage of cells in the ReNcell VM UT cultures, which significantly increased by ~2-fold (*p* = 0.0277) in BDNF-supplemented cultures, resulting in approximately 1% of MAP2+ cells (Figure 5g,h). MAP2 is primarily responsible for stabilising microtubules in neurons [73], suggesting that BDNF supplementation increased neuronal lineage capacity. The percentage of MAP2+ cells in the PDGF-supplemented cultures did not differ from the UT cultures. An examination of changes in glial lineage potential showed that the signal intensity of the astrocyte marker GFAP did not alter significantly following growth factor supplementation (Figure 5i,j), with the oligodendrocyte marker O1 signal intensity maintained in the BDNF-treated cultures and increased significantly in the PDGF-treated cultures (*p* = 0.0225) when compared with UT cultures (Figure 5k,l).

An examination of the changes in the gene expression of neural lineage markers in the ReNcell VM BDNF-supplemented neural inductive cultures identified the astrocyte marker *SLC1A3* (*p* = 0.0048) as the only marker to be significantly downregulated (Figure 6a). This observation suggests that while the proportion of astrocytic cells may have increased, along with increased GFAP expression, the resulting astrocytes were immature in function. In the PDGF-supplemented ReNcell VM neural cultures, a significant downregulation of the astrocyte markers *S100B* (*p* = 0.0071), *SLC1A3* (*p* = 0.0161) and *GFAP* (*p* = 0.0366) was observed (Figure 6c), suggesting the suppression of the astrocyte lineage. Interestingly, exogenous PDGF has previously been shown to promote the glial differentiation of human stem cells in vitro [74,75,76], thus it is likely that the addition of PDGF to the ReNcell VM neural cultures in this study favoured the promotion of oligodendrocyte over astrocyte lineage differentiation.

### 3.9. Growth Factor Supplementation to Neural Cultures Remodelled HS 6-O-Sulfation Sites

Both BDNF and PDGF-BB are HS-binding neuroregulatory growth factors; however, the effect of their exogenous supplementation on the HSPG profile of human NPCs has not been well characterised. The gene expression analysis of HSPG core proteins and biosynthetic enzyme genes in the ReNcell VM neural cultures modulated with BDNF showed a significant downregulation of *GPC6* (*p* = 0.0286) and a significant upregulation of *GPC5* (*p* = 0.0157) when compared with the ReNcell VM UT cultures (Figure 6b). The HS modification enzymes, *HS6ST1* (*p* = 0.0004) and *SULF2* (*p* = 0.0367), were also found to be significantly upregulated. The HS6ST1 enzyme facilitates the 6-*O*-sulfation of HS [77], with SULF2 being an endosulfatase which removes 6-*O*-sulfation [78]. The upregulation of these two genes suggests ongoing modification at the 6-*O*-sulfate sites in response to exogenous BDNF. In the PDGF-modulated cultures, no significant changes in core protein gene expression were observed; however, a significant upregulation of the enzyme genes *HS6ST1* (*p* = 6.19 × 10^−7^) and *HPSE* (*p* = 0.0378) (Figure 6d) was observed. HPSE is an endoglycosidase active during ECM remodelling by cleaving HS [79]. Together with *HS6ST1*, these changes in gene expression suggest that exogenous PDGF-BB resulted in localised ECM remodelling. The WB analysis of HSPGs showed a reduced expression of GPCs, SDC3 and SDC1 core proteins in the growth factor-modulated ReNcell VM (Figure 6e) and ReNcell VM UT cultures (Figure 6f). The analysis of HS content in the BDNF- and PDGF-supplemented cultures showed a similar 10E4 signal intensity to the UT cultures (Figure 6g,h).

## 4. Discussion

The cell surface-bound SDC and GPC HSPGs comprise a core protein with at least one or more HS GAG chains attached. HS GAGs are the result of complex post-translational biosynthethic processes involving a suite of enzymes [80]. HSPGs are highly abundant in the NSC niche with a vast array of functions, including modulating NSC self-renewal, differentiation, lineage commitment and specification [12]. However, many of the functions of HSPGs in human models of neurogenesis are still largely unknown. In this study, three human neural cell lines were explored and characterised to identify the changes specific to astrocytic lineage capacity. We examined these neural cell lines under both basal expansion and inductive culture conditions to assess neural-specific marker expression alongside HSPG profiles, with the goal of developing a simple and reproducible astrocyte model from the commercially available progenitor lines. While neural cell lines are indispensable tools for advancing our understanding of human neurogenesis and cellular differentiation, they lack direct therapeutic applicability [81]. Their value lies in their lineage differentiation potential, including into astrocytes, making them ideal for studying the disruptions in lineage specification that may underlie the dysfunctional or reactive astrocyte states implicated in neurodegenerative diseases [82]. These cell lines offer an accessible, scalable platform for dissecting the molecular decision making processes that govern astrocytic fate—an insight that is ultimately translatable to primary cell and therapeutic models. Given their reproducibility and ease of use, the cell models used in this study provide a powerful framework to construct an HSPG-based molecular roadmap for astrocyte induction and glial lineage commitment.

We showed that at D14, ReNcell CX cells exhibited spontaneous calcium oscillatory activity consistent with astrocytic signalling patterns, whereas differentiated SH-SY5Y cells displayed activity more indicative of neuronal lineages. In contrast, ReNcell VM cultures only exhibited comparable oscillatory patterns under growth factor-supplemented conditions, with PDGF treatment eliciting notably higher Ca^2+^ peak amplitudes. This data underscored the intrinsic heterogeneity of the ReNcell VM population and highlighted the limitations of short-term differentiation protocols in achieving lineage-specific homogeneity. The enhanced oscillatory response to PDGF may reflect an enrichment of glial-biased progenitors within the VM population, with PDGF potentially driving differentiation toward oligodendrocyte-like states.

Previous studies conducted by our group examined hNSC H9 neuronal differentiation in short-term (D15–18) and long-term cultures (D40–60) [15,49]. Similar to SH-SY5Y neuronal cultures, the D15–18 hNSC cultures also expressed the immature neuronal markers TUBB3, *NEFM* and *DCX* with downregulation of the astrocyte marker S100B [14]. Differences in cell morphology were also observed, with SH-SY5Y neuronal cultures displaying cell clustering, which is not characteristic in hNSC H9 neuronal cultures. The SH-SY5Y and hNSC H9 neuronal cultures were examined under different inductive culture conditions, likely influencing differences in the stage of maturation between the two cell types, despite being in lineage differentiation inductive culture conditions for a similar length of time. Neuronal D60 hNSC H9 cultures also showed the significant gene expression of the mature neuronal marker *MAP2*, as well as small cell bodies with long dendritic-like processes, and the formation of neural clusters and inter-cluster connections [15]. Combined, this data indicated that SH-SY5Y neuronal cultures exhibited a phenotype indicative of a stage of maturation between the hNSC H9 D15–18 and D60 time-point cultures.

Interestingly in ReNcell VMs, CD SDC3 was found to be upregulated in spontaneously differentiating cultures, while its MB and TMB forms were observed to be downregulated relative to basal conditions. This suggested that intracellular signalling mediated by CD SDC3 is particularly relevant during early lineage specification. The SDC3 CD interacts with cytoskeletal regulators and signalling proteins and is released via intramembrane proteolysis by the γ-secretase complex—an event that may interface with Notch intracellular signalling, which is known to influence neural development [62]. Increased CD SDC3 levels in spontaneously differentiating ReNcell VM cultures may therefore reflect active Notch signalling—a pathway that must be suppressed for terminal differentiation to proceed [83]. This is further supported by the observed elevated SDC3 expression during brain development in regions undergoing active neurogenesis and downregulated as neurons mature [71]. Furthermore, along with the marked downregulation of CD SDC3 observed in SH-SY5Y-derived neuronal cultures, these cultures also exhibited more defined calcium oscillatory behaviour than ReNcell VM cells. Moreover, in ReNcell VM cultures treated with growth factors, the observed inverse relationship between calcium oscillation amplitude and SDC3 expression further suggested a regulatory role for SDC3 in the balance between progenitor maintenance and lineage commitment, potentially through the Notch pathway. This is further strengthened by the complete absence of both MB and CD SDC3 in astrocyte cultures.

Interestingly, the persistent, albeit reduced, expression of TMB SDC3 in the astrocyte cultures suggested a continued functional role for this isoform in astrocyte physiology. Prior studies have demonstrated that TMB SDC3—particularly its shed ectodomain—can modulate extracellular signalling dynamics, including interactions with growth factors and matrix components [84]. Given the prominent role of astrocytes in secreting HS and PGs, it is plausible that TMB SDC3 remains expressed to support the regulated secretion of these molecules [85,86]. Notably, soluble SDC3 has also been implicated in the internalisation and fibrillation of α-synuclein—an event closely associated with neurodegenerative disease pathogenesis [87,88]. Further suggesting astrocytic preparation for HS secretion, we observed a marked shift in 10E4 staining patterns—from a diffuse distribution to distinct membrane clustering—indicating the localised enrichment of HS. This change coincided with the upregulation of HS biosynthetic enzymes *EXT1*, *EXT2* and *C5-EP*, suggesting the active elongation of HS chains. Interestingly, this also occurred alongside the downregulation of key sulfation enzymes such as *NDST2*, *NDST4* and *HS2ST1*, while 6-*O*-sulfation and related modification enzyme activity was maintained. These selective changes in enzyme expression profiles may reflect a regulatory mechanism favouring dynamic remodelling and functional specificity of HS side chains in differentiated astrocytes.

As indicated previously, the addition of PDGF to ReNcell VM neural inductive cultures appeared to suppress the astrocyte lineage and prompt oligodendrocyte lineages. This was shown by the significant downregulation observed for astrocyte markers *GFAP*, *S100B* and *SLC1A3*, while O1 signal intensity was observed to increase. Interestingly, the expression of HSPGs remained unchanged in +PDGF cultures, apart from the upregulation of *HS6ST1* and *HPSE*, further providing evidence for the importance of 6-*O*-sulfation and HS shedding in glial cells. Notably, HS was also increased in the neuronal SH-SY5Y cells; however, there was a marked reduction in several *N*- and 6-*O*-sulfation enzymes along with increased *SULF2* expression, further suggesting that 6-*O*-sulfation patterns are glial-specific. *SULF2* gene expression was also significantly upregulated in BDNF-treated ReNcell VM cultures, suggesting that BDNF increased the neuronal potential of the heterogenous ReNcell VM cells. In mice, Sulf2 has been shown to be required for neurite outgrowth [89] and neural fate decisions from motor neurons to oligodendrocytes through the Sonic Hedgehog (Shh) signalling pathway [90]. While changes in HS sulfation as reflected by the differential expression of biosynthesis and modification enzymes may play a key role in HSPG-mediated lineage specification, it is essential to validate these findings through functional studies, such as targeted knockdown and enzymatic inhibition. Additionally, employing GAG sequencing techniques would be invaluable to characterise the temporal alterations in HS side chains during both in vitro and in vivo neural cell culture differentiation. We encourage future studies to pursue these approaches to further elucidate the mechanistic roles of HS modifications in neural lineage specification.

Interestingly, BDNF treatment also led to an increased expression of *GPC5* in ReNcell VM cultures. *GPC5* has been shown to promote the proliferation of mouse cerebellar granule cells by acting as a co-receptor for Shh signalling, particularly through its 2-*O*-sulfated HS residues [91]. This upregulation correlated with the observed increase in cell numbers in BDNF-treated cultures, accompanied by decreased *GPC6* expression. These findings suggest a possible functional divergence between the two GPCs, where *GPC5* may support neuronal proliferation or maintenance, while *GPC6* may be more relevant to astrocytic lineage commitment, consistent with its observed elevated expression in ReNcell CX-derived astrocyte cultures. Previous work by Allen et al. (2012) also identified GPC4 and GPC6 as astrocyte-secreted HSPGs involved in promoting synaptogenesis [92], while our previous work further established *GPC4* as a key neural HSPG [15].

Most notably, the expression pattern of GPC2—also known as cerebroglycan—was particularly striking. Similar to SDC3, GPC2 exists in multiple isoforms. It is reported to be expressed in the adult brain and has been identified as a secreted marker of immature neurons in hNSC models [93]. While the core protein of GPC2 is approximately 62 kDa, extensive glycosylation and HS chain modification increases its apparent molecular weight to ~100 kDa on an SDS-PAGE. This highly glycosylated form is enriched in neural tissues, where it plays a critical role in neuronal development, including cell adhesion and neurite outgrowth [94,95,96]. In the present study, we observed a marked upregulation of glycosylated GPC2 in differentiated SH-SY5Y and mixed lineage ReNcell VM cultures. Interestingly, although to a lesser extent, glycosylated GPC2 was also seen to be elevated in ReNcell CX-derived astrocyte cultures. While modest, this expression suggests that the post-translational modification of GPC2 may serve as a biomarker of lineage commitment within NPC-derived systems. These findings are consistent with reports showing the dynamic modulation of GPC2 expression during neurodevelopment, where different isoforms may influence distinct stages of cell maturation [95].

Collectively, these results suggest that the glycosylation of GPC2 enhanced its extracellular signalling capacity, particularly during lineage specification. The differential regulation of PG isoforms, including GPC2, underscored their complex and dynamic role in neural cell lineage fate processes. Future studies incorporating the siRNA-mediated knockdown of GPC2 prior to differentiation could further elucidate its specific regulatory functions and potential as a target in modulating lineage outcomes.

## 5. Limitations

While this study provides valuable insights into the expression profiles of HSPGs and related biosynthetic enzymes during neural lineage specification, several limitations must be acknowledged. Firstly, our findings primarily rely on gene and protein expression analyses, which do not necessarily reflect enzyme or protein functional activity. It is well documented that enzyme activity can significantly differ from expression levels due to various biological processes, such as differences in mRNA stability, translational efficiency, post-translational modifications, protein stability and enzyme turnover rates [97,98,99,100]. Thus, caution must be taken when interpreting our data in terms of functional outcomes.

Secondly, our study did not include direct biochemical analyses of HS GAG structures or sulfation patterns. Such analyses are crucial for confirming the functional implications suggested by changes in gene and protein expression. Variability in HS structure, including sulfation patterns, chain length and epimerisation, necessitates detailed structural analysis using methods such as mass spectrometry or high-performance liquid chromatography (HPLC). These techniques were beyond the scope of the present investigation [11,101]. Consequently, the interpretations linking the observed expression changes directly to the alterations in HS GAG structure should be considered tentative.

Furthermore, the use of immortalised neural cell lines (SH-SY5Y, ReNcell CX and ReNcell VM), while beneficial for reproducibility and ease of handling, presents inherent limitations including reduced physiological relevance when compared with primary neural progenitors with some potential for culture-induced artefacts [11,101]. Thus, generalising our findings to primary neural progenitors or in vivo conditions must also be approached cautiously. Further validation approaches using primary or stem cell-derived neural cells would be required to enhance physiological relevance. As an example, future investigations could incorporate direct functional assays, detailed HS structural analyses and comparative studies with primary neural cells to validate and extend the findings presented in this study.

## 6. Conclusions

This study revealed key insights into the dynamic regulation of HSPGs during neural lineage specification, with a particular emphasis on astrocytic lineage commitment. Although neither of the ReNcell models achieved full maturity or demonstrated homogeneous astrocyte cultures, ReNcell CXs most closely developed an astrocytic phenotype, while PDGF-treated ReNcell VMs exhibited a preferential bias toward an oligodendrocyte lineage. Notably, SDC3 emerged as a lineage-sensitive PG, with its cytoplasmic domain enriched in progenitor-like states and absent in inductive cultures—supporting its role in maintaining neural plasticity. In contrast, the persistence of TMB SDC3 in astrocyte cells suggests a continued role in modulating extracellular signalling and perhaps the priming of the cells for increased HS and PG secretions, as shown by the HS aggregations on the cell membrane. We showed that 6-O sulfation patterns may be relevant for glial lineages, with the PDGF treatment of ReNcell VM cultures suppressing the promotion of more oligodendrocyte-like features, confirming the capacity of the cells to bias glial lineage fate. Intriguingly, GPC2—a glycosylated neural proteoglycan—was upregulated in both the neuronal and astrocytic cultures, indicating its potential as a post-translational biomarker of lineage progression. Together, these findings highlight the distinct isoforms of SDC3 and GPC2 as central modulators of lineage specification in NPC-derived systems and underscore the utility of HSPG profiling in the refinement of human astrocyte model development.

## Figures and Tables

**Figure 1 cells-14-01158-f001:**
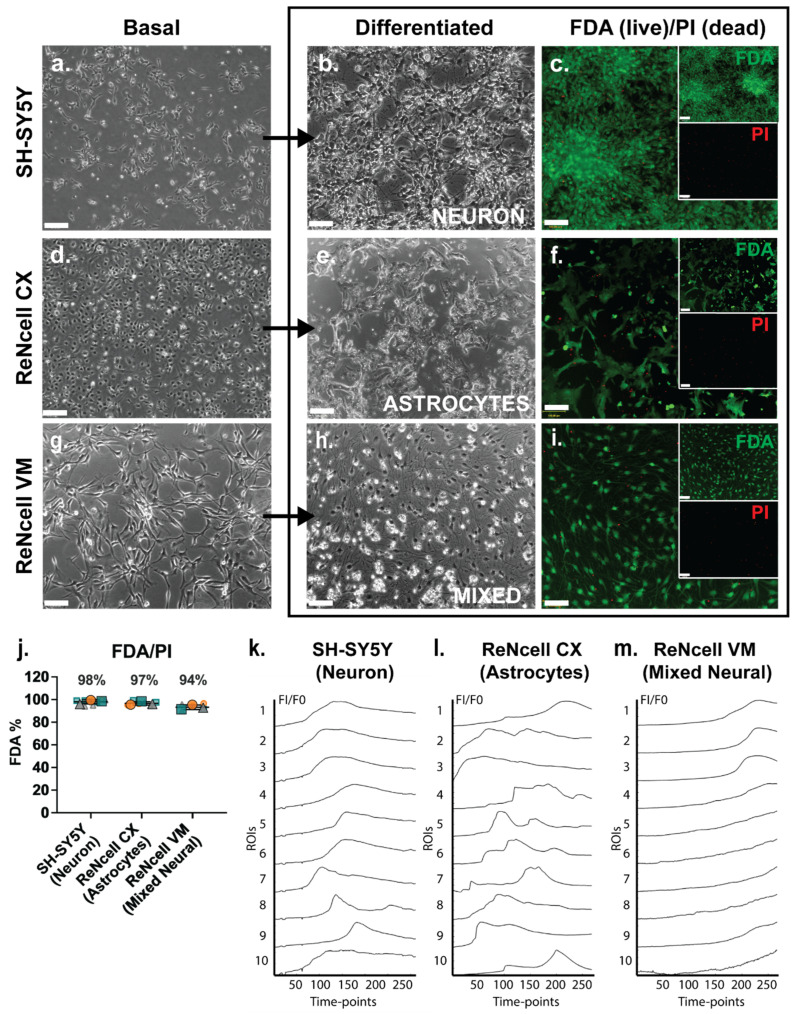
Lineage-specific differentiated human neural cell lines exhibited high viability and spontaneous calcium activity. Phase contrast images of basal, differentiated and FDA/PI-stained (**a**–**c**) SH-SY5Y human neuroblastoma cell line-derived neuronal cultures at D18 (D18), (**d**–**f**) ReNcell CX human neural progenitor cell line-derived astrocyte cultures at D14 and (**g**–**i**) ReNcell VM human neural progenitor cell line-derived mixed neural cultures at D14, following spontaneous differentiation via withdrawing the growth factors fibroblast growth factor 2 (FGF-2) and EGF. Inset: individual FDA (green) and PI (red) channels. All images are 10× magnification, scale bar = 130 μm. (**j**) Signal intensity of FDA (= live) and PI (= dead) staining of neural cell line differentiated cultures. Error bar = SD. Examination of spontaneous calcium activity using Fluo-4 Ca^2+^ indicator dye in (**k**) SH-SY5Y neuronal cultures, (**l**) ReNcell CX astrocyte cultures and (**m**) ReNcell VM mixed neural cultures. Neural cultures were incubated with Fluo-4 calcium reagent and imaged for 200 s with 0.8 s intervals (detected using the FITC filter). Ca^2+^ signals (FI/F0) were calculated from the fluorescence intensity (FI) of 10 regions of interests (ROIs) manually selected within each culture condition across all 251 timepoints and normalised to the fluorescence intensity of timepoint 1 (F0).

**Figure 2 cells-14-01158-f002:**
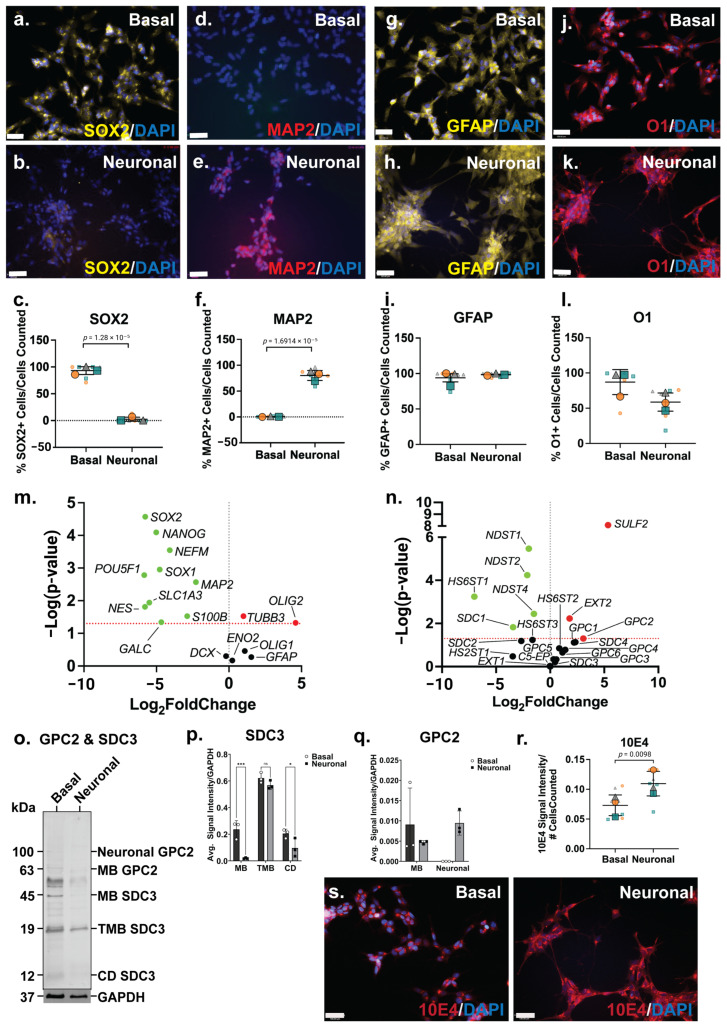
Phenotype of SH-SY5Y neuronal cultures and its heparan sulfate proteoglycan (HSPG) profile. Immunocytochemistry analysis of (**a**,**b**) neural stem cell (NSC) marker SRY-box 2 (SOX2), (**d**,**e**) mature neuron marker microtubule-associated protein 2 (MAP2), (**g**,**h**) astrocyte marker glial fibrillary acidic protein (GFAP) and (**j**,**k**) oligodendrocyte marker O1, in basal and differentiated SH-SY5Y neuronal cultures. Images taken at 20× magnification, scale bar = 130 μm. Quantification of percentages of (**c**) SOX2+, (**f**) MAP2+, (**i**) GFAP+ and (**l**) O1+ cells within basal and differentiated SH-SY5Y neuronal cultures. Error bar = SD; statistical significance detected by Student’s *t* test denoted ns = no significance, * *p* < 0.05, ** *p* < 0.01, *** *p* < 0.001, **** *p* < 0.0001. Volcano plots depicting changes in gene expression level of (**m**) neural genes and (**n**) HSPG-related genes. Significance cut-off at *p* < 0.05. Data points in green are significant and have <0-fold change, the red data points are significant and have >0-fold change and the black data points are not significant. (**o**) Western blot analysis of polyclonal glypican-2 (GPC2, including membrane-bound (MB) GPC2 63 kDa and glycosylated or neuronal GPC2 100 kDa) and syndecan-3 (SDC3; including membrane-bound (MB) 45 kDa, truncated membrane-bound (TMB) 19 kDA and cytoplasmic domain (CD) 12 kDA) were observed in SH-SY5Y basal and neuronal differentiated cultures. (**p**) Optical density of MB and neural SDC3 protein bands quantitated using Image Lab v6.1.0 build 7 (BioRad) and target signals were normalised to the loading control GAPDH. (**q**) Optical density of MB, TMB and CD SDC3 protein bands quantitated using Image Lab (BioRad) and target signals were normalised to the loading control GAPDH. (**r**) Comparison of localization of heparan sulfate (HS) glycosaminoglycan (GAG) chains in basal and differentiated SH-SY5Y neuronal cultures through immunostaining analysis using Pan-HS 10E4 antibody. Images taken at 20× magnification, scale bar = 130 μm. (**s**) Signal intensity of 10E4 in SH-SY5Y basal and neuronal cultures normalised to the number of cells (DAPI). All error bars = SD; statistical significance detected by unpaired *t* test with Welch’s correction and two-way ANOVA.

**Figure 3 cells-14-01158-f003:**
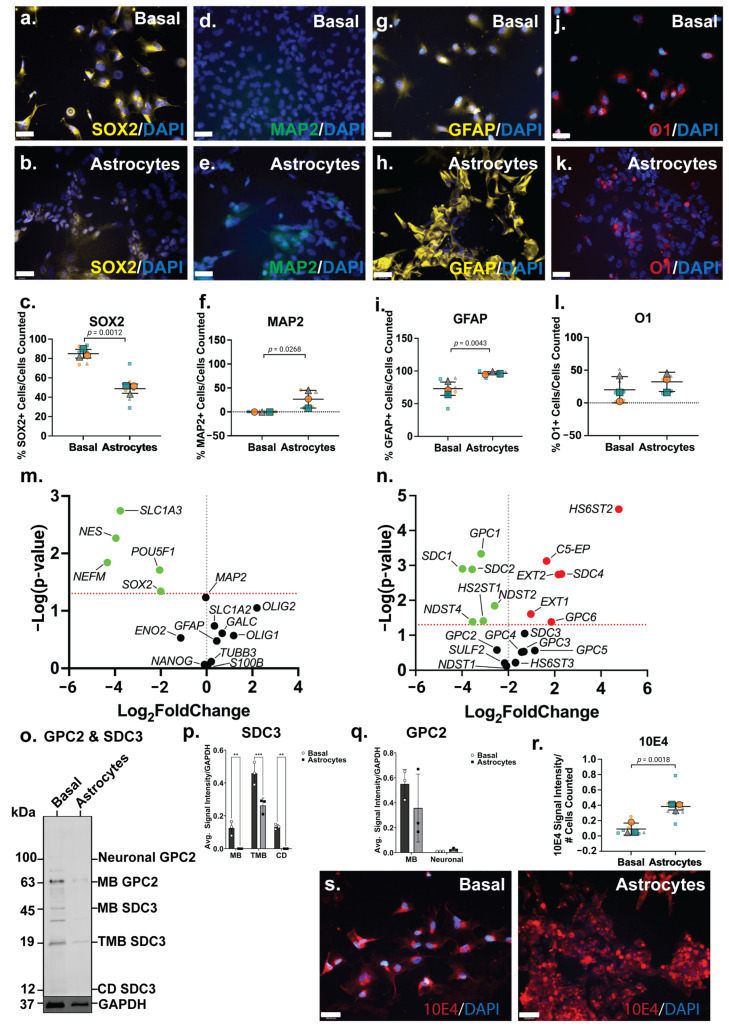
Phenotype of ReNcell CX astrocyte cultures and its heparan sulfate proteoglycan (HSPG) profile. Immunostaining analysis of (**a**,**b**) neural stem cell (NSC) marker SRY-box 2 (SOX2), (**d**,**e**) mature neuron marker microtubule-associated protein 2 (MAP2), (**g**,**h**) astrocyte marker glial fibrillary acidic protein (GFAP) and (**j**,**k**) oligodendrocyte marker O1, in basal and differentiated ReNcell CX astrocyte cultures. Images taken at 20× magnification, scale bar = 130 μm. Quantification of percentages of (**c**) SOX2+, (**f**) MAP2+, (**i**) GFAP+ and (**l**) O1+ cells within basal and differentiated ReNcell CX astrocyte cultures. Error bar = SD; statistical significance detected by Student’s *t* test denoted ns = no significance, * *p* < 0.05, ** *p* < 0.01, *** *p* < 0.001, **** *p* < 0.0001. Volcano plots depicting changes in gene expression level of (**m**) neural genes and (**n**) HSPG-related genes. Significance cut-off at *p* < 0.05. Data points in green are significant and have <0-fold change, the red data points are significant and have >0-fold change and the black data points are not significant. (**o**) Western blot analysis of polyclonal glypican-2 (GPC2, including membrane-bound (MB) GPC2 63 kDa and glycosylated or neuronal GPC2 100 kDa) and syndecan-3 (SDC3; including membrane-bound (MB) 45 kDa, truncated membrane-bound (TMB) 19 kDA and cytoplasmic domain (CD) 12 kDA) were observed in ReNcell CX basal and astrocyte differentiated cultures. (**p**) Optical density of MB and neural SDC3 protein bands quantitated using Image Lab (BioRad) and target signals were normalised to the loading control GAPDH. (**q**) Optical density of MB, TMB and CD SDC3 protein bands quantitated using Image Lab (BioRad) and target signals were normalised to the loading control GAPDH. (**r**) Comparison of localization of heparan sulfate (HS) glycosaminoglycan (GAG) chains in basal and differentiated ReNcell CX astrocyte cultures using Pan-HS 10E4 antibody. Images taken at 20× magnification, scale bar =130 μm. (**s**) Signal intensity of 10E4 in ReNcell CX basal and astrocyte cultures normalised to the number of cells (DAPI). All error bars = SD; statistical significance detected by unpaired *t* test with Welch’s correction and two-way ANOVA.

**Figure 4 cells-14-01158-f004:**
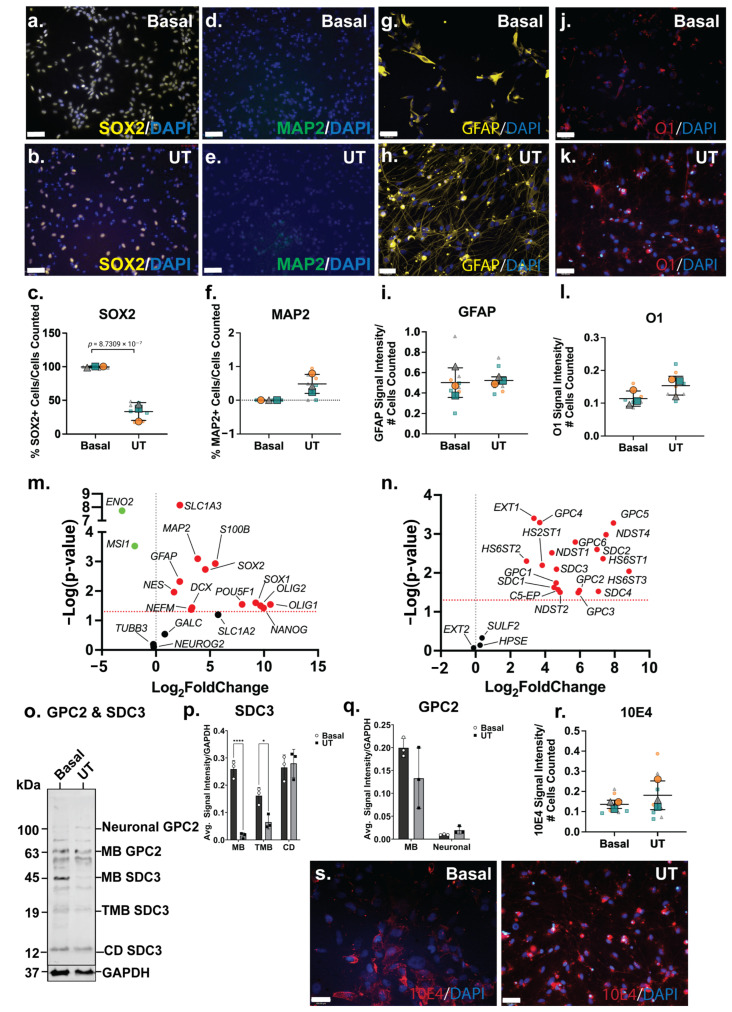
Phenotype of ReNcell VM mixed neural cultures and its heparan sulfate proteoglycan (HSPG) profile. Immunostaining analysis of (**a**,**b**) neural stem cell (NSC) marker SRY-box 2 (SOX2), (**d**,**e**) mature neuron marker microtubule-associated protein 2 (MAP2), (**g**,**h**) astrocyte marker glial fibrillary acidic protein (GFAP) and (**j**,**k**) oligodendrocyte marker O1, in basal and differentiated ReNcell VM mixed neural cultures. Images taken at 20× magnification, scale bar = 130 μm. Quantification of percentages of (**c**) SOX2+, (**f**) MAP2+, (**i**) GFAP+ and (**l**) O1+ cells within basal and differentiated ReNcell CX astrocyte cultures. Error bar = SD; statistical significance detected by Student’s *t* test denoted ns = no significance, * *p* < 0.05, ** *p* < 0.01, *** *p* < 0.001, **** *p* < 0.0001. Volcano plots depicting changes in gene expression level of (**m**) neural genes and (**n**) HSPG-related genes. Significance cut-off at *p* < 0.05. Data points in green are significant and have <0-fold change, the red data points are significant and have >0-fold change and the black data points are not significant. (**o**) Western blot analysis of polyclonal glypican-2 (GPC2, including membrane-bound (MB) GPC2 63 kDa and glycosylated or neuronal GPC2 100 kDa) and syndecan-3 (SDC3; including membrane-bound (MB) 45 kDa, truncated membrane-bound (TMB) 19 kDA and cytoplasmic domain (CD) 12 kDA) were observed in ReNcell VM basal and untreated differentiated cultures. (**p**) Optical density of MB and neural SDC3 protein bands quantitated using Image Lab (BioRad) and target signals were normalised to the loading control GAPDH. (**q**) Optical density of MB, TMB and CD SDC3 protein bands quantitated using Image Lab (BioRad) and target signals were normalised to the loading control GAPDH. (**r**) Comparison of localization of heparan sulfate (HS) glycosaminoglycan (GAG) chains in basal and differentiated SH-SY5Y neuronal cultures through immunostaining analysis using Pan-HS 10E4 antibody. Images taken at 20× magnification, scale bar = 130 μm. (**s**) Signal intensity of 10E4 in SH-SY5Y basal and neuronal cultures normalised to the number of cells (DAPI). All error bars = SD; statistical significance detected by unpaired *t* test with Welch’s correction and two-way ANOVA.

**Figure 5 cells-14-01158-f005:**
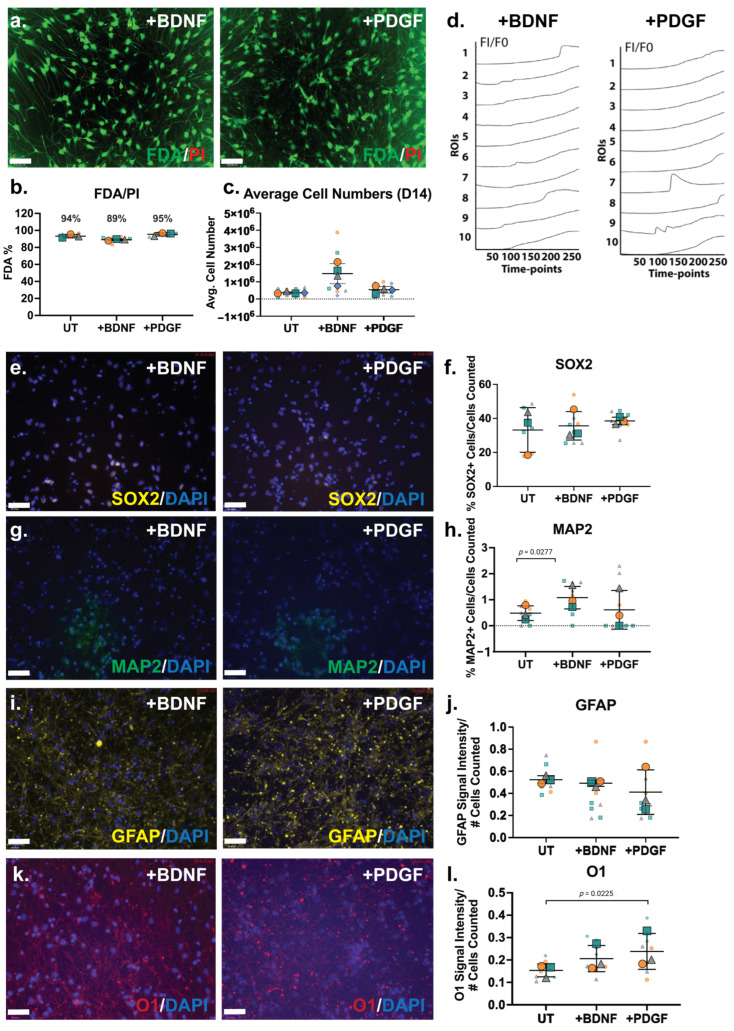
Characterisation of ReNcell VM neural differentiation D14 cultures modulated with HS-binding neurogenic growth factors BDNF and PDGF-BB. (**a**) Cell viability analysis of ReNcell VM +BDNF and +PDGF D14 differentiation cultures by FDA/PI staining. FDA (live cells) = green, PI (dead cells) = red. Scale bar = 130 μm. (**b**) FDA and PI signal intensities were quantitated and presented as a bar graph. Error bars = SEM. (**c**) Average cell numbers of untreated (UT), +BDNF and +PDGF ReNcell VM neural cultures at D14. Error bars = SD. (**d**) Fluo-4 calcium assay signal quantitation (FI/F0) of 10 manually selected regions of interests (ROIs) in ReNcell VM +BDNF and +PDGF modulated differentiation cultures at D14, measured for 200 s with 0.8 s intervals = 251 timepoints. FI = fluorescence intensity, F0 = fluorescence intensity of time-point 1. Immunocytochemistry analysis of (**e**) neural stem cell (NSC) marker SOX2, (**f**) percentage of SOX2+ cell quantitation, (**g**) mature neuron marker MAP2, (**h**) percentage of MAP2+ cells quantitation, (**i**) astrocyte marker GFAP, (**j**) quantitation of GFAP signal intensity normalised to cell number (DAPI), (**k**) oligodendrocyte marker O1 in ReNcell VM +BDNF and +PDGF supplemented cultures and (**l**) Quantitation of O1 signal intensity normalised to cell number (DAPI). Images taken at 20X magnification, scale bar = 130 μm. All error bars = SD; statistical significance detected by unpaired *t* test with Welch’s correction.

**Figure 6 cells-14-01158-f006:**
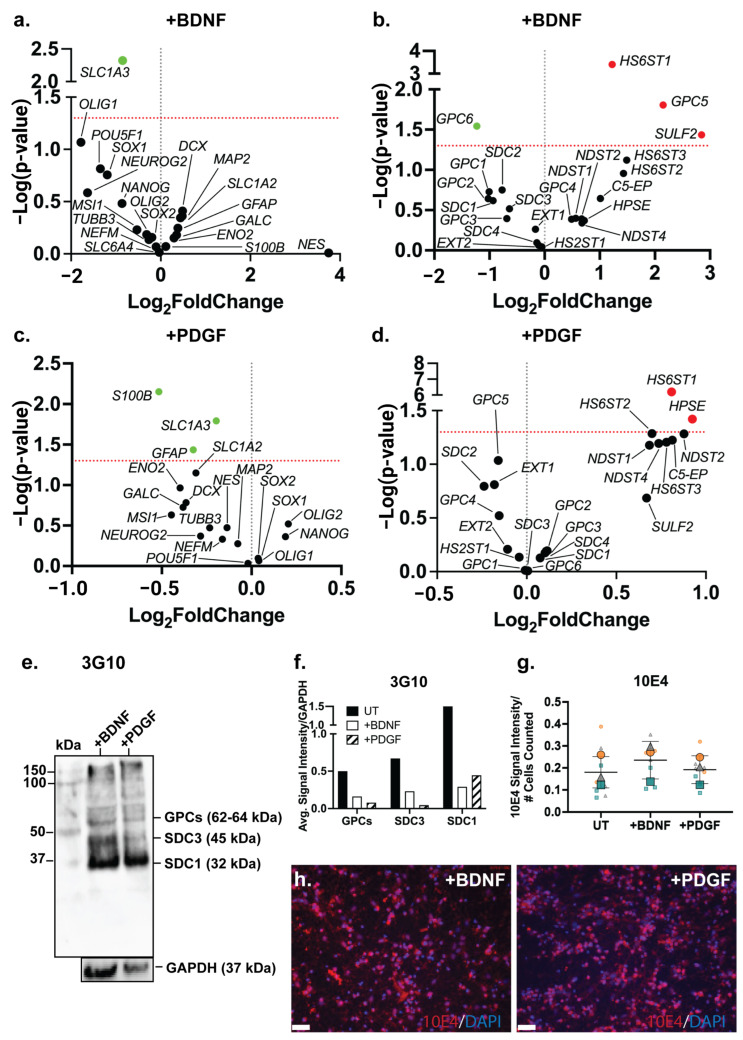
Neural marker gene expression and HSPG profile of ReNcell VM mixed neural cultures supplemented with +BDNF and +PDGF. Volcano plots depicting gene expression changes in (**a**) neural markers and (**c**) HSPG-related genes in ReNcell VM neural cultures supplemented with 10 ng/mL brain-derived neurotrophic factor (BDNF) in comparison to ReNcell VM untreated (UT) cultures. Volcano plots depicting gene expression changes in (**b**) neural markers and (**d**) HSPG-related genes in ReNcell VM neural cultures supplemented with 10 ng/mL platelet-derived growth factor (PDGF) compared to ReNcell VM untreated (UT) cultures. (**e**) Western blot analysis of HSPG core proteins using the pan-HS 3G10 antibody. Glypicans (GPCs, ~62–64 kDa), syndecan-3 (SDC3; 45 kDa) and syndecan-1 (SDC1; 32 kDa) were observed in ReNcell VM neural cultures supplemented with +BDNF and +PDGF, with GAPDH (37 kDa) as the loading control. (**f**) Quantitation of optical density of 3G10 protein bands, target signals were normalised to the loading control GAPDH. (**g**) Localization of HS GAG chains in ReNcell VM neural cultures supplemented with +BDNF and +PDGF through immunostaining analysis using Pan-HS 10E4 antibody. Images taken at 20X magnification, scale bar = 130 μm. (**h**) Signal intensity of 10E4 in ReNcell VM UT, +BDNF and +PDGF neural cultures, normalised to the number of cells (DAPI). All error bars = SD.

## Data Availability

The datasets generated and analysed during the current study are available from the corresponding author on reasonable request.

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
