# Peer review of "Heparan Sulfate Proteoglycans as Potential Markers for In Vitro Human Neural Lineage Specification"

_cells, 2025, doi:10.3390/cells14151158_

Round 1

Reviewer 1 Report

Comments and Suggestions for Authors

This manuscript is one in a collection from the research group that focuses on studying HSPG changes in developing neuronal systems.  As such, the authors have significant experience dealing with this issue.  Here, they present work with several new cell lines that include SH-SY5Y neuroblastoma cells as a control, and two commercial lines that can be differentiated in culture.  They provide evidence for differentiation and use some previous markers for astrocytes and neuronal lineages.  I find the title misleading as the data, with some limitations noted below, do support changes in HSPGs with the differentiation process.  However, there are no mechanistic studies that support HSPG changes mediating the differentiation as the title implies.  Rather, the data are a generally nice representation of correlative studies.  In fact, in the text the authors talk about these HSPG changes as new markers, and might be involved in the differentiation process.  The title should be changed to reflect the identification of HSPG markers rather than indication of mechanistic relationships.

Specific comments on data on HSPGs in Figure 2 include.  First, here and throughout, the authors reference primers that were published previously.  While okay, it should be that the reference where each primer is identified by sequence should be given rather than a cluster of references, which could make an interested user chase through many references to find the right primer.  Alternatively, the primers could be listed in a table in methods, with references to where they were first used.

Second, the authors explain in the Methods that initial identification of HSPGs is done with an antibody that recognizes "HS stubs" and then use gene cards information to figure out what molecular weight they should observe.  The use of gene card numbers rather than confirming the identification of each band they find as matching the HSPG protein in question is inappropriate.  Among concerns, many people have shown that syndecan protein cores dimerize and are often not separated by standard reducing conditions for SDS gel electrophoresis.  Therefore, it is actually more common that dimers for some of these syndecan cores are identified.  That makes it much harder to just identify cores by molecular weight, plus there is no indication of antibody verification for each of these bands.

In panel o, there are several problems.  First, the panel showing red on black is very difficult to see (the original images in the supplementary data are somewhat easier to see, but raise additional questions).  Can you please make an inverted image to show the bands more clearly? Looking at the Methods, it appears that these western blots were not stained with the stub antibody. Does that mean that SDC3 and GPC2 antibodies were mixed together and used?  That is really unusual, and should at least be stated in the legend the first time that is done. On top of this, the authors mention specific degradation products of GPC2, in particular.  I don't see confirmation referenced or data in their supplemental work showing that the antibodies they are using recognize all of the degradation bands that they quantify.  Further, it seems as though all of the repeats are run on one blot. This is not really clear in either the figure legend, or methods. Are they technical and/or experimental repeats. Do the authors think that the changes in HS chain production enzymes are critical, if so, it would be appropriate to try and address this mechanistically as none of the data outside of the volcano plot data really provide any information.

Figures 3 and 4 have the same HSPG western data problems as identified in Figure 2.  All of these need to be easier to see, more carefully explained, and evidence beyond a reference for a band that size in someone else's work indicating that the staining is actually seeing degradation products the authors identify.  

The ReNCell VM line appears very complex, and the authors spend a lot of effort trying to make it fit with the other cell lines, finding HSPG changes.  Does this mixed line really provide useful evidence regarding the HSPGs as potential markers? since that seems to be the real outcome of this study.  The stub (3G10) antibody staining shown in Figure 6e is not sufficiently clear to identify HSPG core protein differences.  For starters, the control staining is very different between the two induction methods, and no UT lane is shown on this blot.  This is not sufficient evidence from which to produce a graph.  

Previous studies from the lab on GPC1 and GPC4 in these cells could be better referenced here for comparison (likely in the discussion).  This would make the paper a better reference for someone interested in these HSPGs as markers. 

Comments on the Quality of English Language

General comment on English.  Mostly, it is okay.  However, a number of subject verb disagreements were irritating.

What does “medication” of sulfation sites in section 3.3 mean?  Increased GAG content I get, how does the image really show increased GPC as opposed to a mix of GPC and SDC4 (slightly increased, though barely below rather than barely above your cutoff) when you are employing an antibody against HS chains, generally?

Author Response

This manuscript is one in a collection from the research group that focuses on studying HSPG changes in developing neuronal systems.  As such, the authors have significant experience dealing with this issue.  Here, they present work with several new cell lines that include SH-SY5Y neuroblastoma cells as a control, and two commercial lines that can be differentiated in culture.  They provide evidence for differentiation and use some previous markers for astrocytes and neuronal lineages.  I find the title misleading as the data, with some limitations noted below, do support changes in HSPGs with the differentiation process.  However, there are no mechanistic studies that support HSPG changes mediating the differentiation as the title implies.  Rather, the data are a generally nice representation of correlative studies.  In fact, in the text the authors talk about these HSPG changes as new markers and might be involved in the differentiation process.  The title should be changed to reflect the identification of HSPG markers rather than indication of mechanistic relationships.

We thank the Reviewer for these supportive and constructive comments. We agree with the suggestion regarding the title and have now revised it to read: “Heparan sulfate proteoglycans as potential markers for in vitro human neural lineage specification.” This change removes the mechanistic implication by replacing “mediators” with “potential markers,” better reflecting the correlative nature of our findings. Additionally, we have broadened “neuronal lineage specification” to “neural lineage specification” to more accurately represent the inclusion of both astrocytic and neuronal differentiation within the manuscript. We also acknowledge the importance of exploring mechanistic roles in future studies and consider this work an initial step in characterising lineage-specific HSPG expression in human cell models.

Specific comments on data on HSPGs in Figure 2 include.  First, here and throughout, the authors reference primers that were published previously.  While okay, it should be that the reference where each primer is identified by sequence should be given rather than a cluster of references, which could make an interested user chase through many references to find the right primer.  Alternatively, the primers could be listed in a table in methods, with references to where they were first used.

We again thank the Reviewer for this suggestion. We have now created a comprehensive table listing all primers used in this study, along with their sequences (Methods Section 2.4. Total RNA Isolation, cDNA synthesis and Q-PCR lines 943-944: Table S1: Primer sequences of all relevant QPCR sets including their RefSeq identification codes for which the primers were designed). This has now been added as Supplementary Table 1.

Second, the authors explain in the Methods that initial identification of HSPGs is done with an antibody that recognizes "HS stubs" and then use gene cards information to figure out what molecular weight they should observe.  The use of gene card numbers rather than confirming the identification of each band they find as matching the HSPG protein in question is inappropriate.  Among concerns, many people have shown that syndecan protein cores dimerize and are often not separated by standard reducing conditions for SDS gel electrophoresis.  Therefore, it is actually more common that dimers for some of these syndecan cores are identified.  That makes it much harder to just identify cores by molecular weight, plus there is no indication of antibody verification for each of these bands.

We thank the Reviewer for the opportunity to clarify this important point. The antibody used for HS stub detection in this was the 3G10 antibody, which we acknowledge is not specific to individual HSPG core proteins. We agree that relying solely on molecular weight is a limitation of HSPG detection. In response, we have now added the following clarification to the Methods section (lines 309–312): “While reliance on molecular weight alone to identify specific core proteins recognised via HS stub detection is a limitation of this approach, the antibody provides a putative range of expected bands consistent with various HSPGs. However, confirmation of individual core protein identities would require validation using protein-specific antibodies.”

In panel o, there are several problems.  First, the panel showing red on black is very difficult to see (the original images in the supplementary data are somewhat easier to see, but raise additional questions).  Can you please make an inverted image to show the bands more clearly?

For ease of visualisation, these images have now been inverted back to black and white.

Looking at the Methods, it appears that these western blots were not stained with the stub antibody. Does that mean that SDC3 and GPC2 antibodies were mixed together and used?  That is really unusual and should at least be stated in the legend the first time that is done.

We again thank the Reviewer for the opportunity to clarify this. In this instance, SDC3 (~45 kDa) and GPC2 (~63 kDa) were detected on the same blot due to the clear separation in their molecular weights, which allowed for simultaneous probing. We agree that this should be stated explicitly and have now clarified this in the Methods section (lines 338–339) with the following text:

“Where appropriate, proteins with non-overlapping molecular weights were probed on the same blot.”

On top of this, the authors mention specific degradation products of GPC2, in particular.  I don't see confirmation referenced or data in their supplemental work showing that the antibodies they are using recognize all of the degradation bands that they quantify.  Further, it seems as though all of the repeats are run on one blot. This is not really clear in either the figure legend, or methods. Are they technical and/or experimental repeats.

We again thank the Reviewer for the opportunity to clarify this. The specific points raised are clarified below followed by relevant details including extensive literature supporting the use of these antibodies and our findings:

  1. Antibody Validation and Band Recognition for GPC2 and SDC3

We appreciate the Reviewer comments regarding the specificity of the antibodies used and the recognition of degradation products in our Western blot analysis. We confirm that the antibodies employed for both GPC2 and SDC3 have been validated to detect multiple physiologically relevant forms of the target proteins, including full-length isoforms and proteolytic fragments.

GPC2 Antibody (Thermo Fisher Scientific, PA5-115299):

This rabbit polyclonal antibody is validated to detect endogenous levels of total Glypican-2 (GPC2), a cell surface heparan sulfate proteoglycan. The antibody recognises both the membrane-bound and neuronal isoforms of GPC2, regardless of glycosylation status or post-translational processing. The presence of multiple bands in our Western blot (including glycanated and potentially cleaved GPC2 fragments) is consistent with previous reports in neuroblastoma and neuronal tissue, where such isoforms have been well documented.

SDC3 Antibody (Abcam, ab63932):

This rabbit polyclonal antibody targets a synthetic peptide within the C-terminal region of human SDC3 and is validated for Western blotting. It detects full-length membrane-bound SDC3 (~45–46 kDa), as well as truncated and cytoplasmic domain-containing fragments, which result from known proteolytic processing of syndecans. These fragments have been observed and validated in multiple publications and reflect physiologically relevant states of the protein.

  1. Supporting Evidence from Literature and Antibody Data Sheets

The observations made in this study and others of multiple bands in Western blots is not indicative of antibody non-specificity but rather reflects the biological complexity of both GPC2 and SDC3. These proteins undergo extensive post-translational modifications, shedding, and intracellular cleavage events that yield a range of detectable fragments. Importantly, the antibodies used in our study target epitopes retained across these fragments, allowing their specific detection.

Protein

GPC2

SDC3

Antibody

Polyclonal

Polyclonal

Catalogue Number

PA-115299

 (ThermoFisher Scientific)

ab63932 (Abcam)

Application

Western Blot

Western Blot

Epitope/Immunogen

Total GPC2 (polyclonal epitope recognition)

Synthetic peptide within human SDC3 (C-terminal domain)

Validated Bands

Multiple forms, including membrane-bound and neuronal isoforms

Full-length, truncated, cytoplasmic fragments

Key References

https://pmc.ncbi.nlm.nih.gov/articles/PMC5600520/

https://pmc.ncbi.nlm.nih.gov/articles/PMC8324494/

https://pmc.ncbi.nlm.nih.gov/articles/PMC5404329/

https://pmc.ncbi.nlm.nih.gov/articles/PMC4791038/

https://pmc.ncbi.nlm.nih.gov/articles/PMC4215216/

https://www.abcepta.com/products/AP14689b-SDC3-Antibody-C-term

  1. Clarification on Replicates: Biological vs. Technical

We acknowledge the need to clarify the nature of replicates in our Western blot experiments.

To clarify, biological triplicates were used throughout the study. These refer to independently prepared samples obtained from separate cell culture experiments conducted on different days, ensuring reproducibility across biological conditions.

For each Western blot presented, the bands shown represent biological replicates, unless otherwise specified. We agree that this information should be stated more clearly within the manuscript. Accordingly, the figure legends and methods sections have now been revised to explicitly distinguish between biological and technical replicates, in alignment with best practices in reporting reproducibility (Line 287-292).

  1. Addressing the Concern on Degradation Product Quantification

The quantification of degradation products for GPC2 and SDC3 was performed based on clearly distinguishable bands observed in the Western blot, each corresponding to known molecular weights of their respective isoforms or fragments. As outlined above, the antibodies used have been validated to recognise both full-length proteins and biologically relevant degradation fragments. These bands were not artefactual or non-specific but instead represent cleavage products that retain the immunogenic epitope targeted by the antibodies.

We have now included the following in the revised manuscript:

  • Antibody catalogue numbers, host species, and targeted epitopes
  • Justification for multiple band recognition based on published validation data and literature
  • Supplementary references confirming the molecular weights and identities of relevant GPC2 and SDC3 fragments

Do the authors think that the changes in HS chain production enzymes are critical, if so, it would be appropriate to try and address this mechanistically as none of the data outside of the volcano plot data really provide any information.

We appreciate this important point. We agree that HS chain synthesis and sulfation enzymes are likely to play a significant role in mediating lineage specification in the models presented in this study. However, mechanistic validation would require targeted functional studies (e.g., gene knockdown or enzymatic inhibition), which are beyond the scope of this characterisation-focused manuscript. We have now clarified this in the Discussion section and highlighted the value of our data as a foundation for future studies to explore the functional relevance of specific HS biosynthetic enzymes and modifications during neural differentiation. To clarify this within our discussion, we have included the following (Page 25, lines 881-889):

 “While changes in HS sulfation as reflected by the differential expression of biosynthesis and modification enzymes may play a key role in HSPG-mediated lineage specification, it is essential to validate these findings through functional studies, such as targeted knockdown and enzymatic inhibition. Additionally, employing GAG sequencing techniques would be invaluable to characterise the temporal alterations in HS side chains during both in vitro and in vivo neural cell culture differentiation. We encourage future studies to pursue these approaches to further elucidate the mechanistic roles of HS modifications in neural lineage specification.”

Figures 3 and 4 have the same HSPG western data problems as identified in Figure 2.  All of these need to be easier to see, more carefully explained, and evidence beyond a reference for a band that size in someone else's work indicating that the staining is actually seeing degradation products the authors identify.  

We thank the Reviewer. As identified previously, we have now inverted these images back to black and white for ease of visualisation.

The ReNCell VM line appears very complex, and the authors spend a lot of effort trying to make it fit with the other cell lines, finding HSPG changes.  Does this mixed line really provide useful evidence regarding the HSPGs as potential markers? since that seems to be the real outcome of this study. 

We agree with the Reviewer that that the ReNcell VM line is a complex and heterogeneous model, but we view this as a strength rather than a limitation. Its cellular diversity (including neural progenitors with distinct lineage bias) more closely reflects the in vivo neurogenic niche, where such heterogeneity is biologically relevant. In contrast, the ReNcell CX line offers a more homogeneous, monoclonal system that likely represents a singular lineage trajectory. This contrast allows us to examine how HSPG-related changes observed in a controlled, lineage-specific environment (ReNcell CX) translate to a more complex, mixed population (ReNcell VM). While the heterogeneity of ReNcell VMs may obscure direct associations between specific HSPG patterns and individual lineages, it provides valuable insight into how HSPG modifications might influence broader neurodevelopmental outcomes, such as progenitor maintenance with differentiation within a more physiologically relevant context. To clarify this within the introduction, we have added the following from lines 131-138:

“While ReNcell CX cells offer a relatively uniform and controlled model of neural differentiation, the inherent heterogeneity of ReNcell VM cultures more closely reflects the cellular diversity present in the neurogenic niches of the developing and adult brain. As such, the ReNcell VM line serves as a valuable model for studying molecular dynamics, such as HSPG expression changes, in a physiologically relevant and complex environment. The use of both these ReNcell models, allows the exploration of the influence of lineage-specific HSPGs on progenitor maintenance and neural differentiation in mixed populations.”

The stub (3G10) antibody staining shown in Figure 6e is not sufficiently clear to identify HSPG core protein differences.  For starters, the control staining is very different between the two induction methods, and no UT lane is shown on this blot.  This is not sufficient evidence from which to produce a graph.  

As per our previous response, we have now included this limitation within the Methods section. Additionally, we have updated the Figure 6 legend to state:

“Specific HSPGs were putatively identified based on observed band sizes that were consistent with the expected molecular weights of known HSPG core proteins.”

We have also now included the UT condition in Supplementary Figure S3: Western blot detection of heparan sulfate proteoglycans in basal versus untreated ReNcell VM cultures using the 3G10 neo-epitope (line 924-925), as requested. However, to clarify, our decision to focus on PDGF- versus BDNF-treated samples in Figure 6 was intentional, as the aim of the Figure was to directly compare the effects of these two differentiation conditions as opposed to their difference from UT samples.

Figure S3: Western blot detection of heparan sulfate proteoglycans in basal versus untreated ReNcell VM cultures using the 3G10 neo-epitope. Western blot analysis of ReNcell VM cells cultured under basal and untreated (UT) conditions following heparitinase digestion and probing with the anti-3G10 antibody revealed distinct bands corresponding to major heparan sulfate proteoglycan (HSPG) core proteins. Bands were detected for glypicans (GPCs, ~62–64 kDa), syndecan 3 (SDC3; 45 kDa), and syndecan 1 (SDC1; 32 kDa), with increased intensity in the UT condition. GAPDH (37 kDa) was used as a loading control. These findings indicate altered HSPG expression profiles between basal and UT conditions, highlighting differential regulation of proteoglycan subtypes during early neural induction.

Previous studies from the lab on GPC1 and GPC4 in these cells could be better referenced here for comparison (likely in the discussion).  This would make the paper a better reference for someone interested in these HSPGs as markers. 

We thank the Reviewer for this helpful suggestion. Our previous work on GPC1 and GPC4 has been referenced at multiple points throughout the manuscript (e.g., lines 166, 168, 791, 876), including in the Introduction, Methods, and Discussion. However, within the context of the discussion we have primarily discussed GPC5 and GPC6, which were not the focus of our previous study. We have now included more details of our previous study within the introduction (lines 92-105).

General comment on English.  Mostly, it is okay.  However, a number of subject verb disagreements were irritating.

Thank you for this observation. We have now carefully reviewed the manuscript and corrected instances of subject–verb disagreements as well as other grammatical errors, hopefully improving overall readability. As an example, we have now corrected:

Lines 54-55: “This is likely as a result of much of what is known about..” to “This is likely because what is known about…”

What does “medication” of sulfation sites in section 3.3 mean? 

We thank the Reviewer, this was a typographical error, and we have now corrected it to “mediation.”

Increased GAG content I get, how does the image really show increased GPC as opposed to a mix of GPC and SDC4 (slightly increased, though barely below rather than barely above your cutoff) when you are employing an antibody against HS chains, generally?

We thank the Reviewer for the opportunity to clarify these in more detail than already provided above. We acknowledge that antibodies targeting heparan sulfate (HS) chains detect the glycosaminoglycan moieties present on multiple heparan sulfate proteoglycans (HSPGs), including GPC2, SDC3, and SDC4. As such, HS chain detection alone does not permit attribution of signal to a specific core protein.

To address this, our quantification of GPC2 and SDC3 was performed using core protein-specific antibodies, not anti-HS chain antibodies. Specifically:

  • GPC2 was detected using ThermoFisher Scientific PA5-115299, a validated polyclonal antibody recognising endogenous levels of total GPC2, including both membrane-bound and neuronal isoforms.
  • SDC3 was detected using Abcam ab63932, a rabbit polyclonal antibody targeting a synthetic peptide within the cytoplasmic domain of human SDC3, capable of identifying full-length, truncated, and cytoplasmic fragments.

These antibodies recognise distinct protein epitopes that are independent of HS chains and do not cross-react with other proteoglycans such as SDC4. This specificity is further supported by the observation of unique and non-overlapping banding patterns for GPC2 and SDC3 on our Western blots and is supported by the literature summarised earlier.

To minimise inter-blot variability and facilitate accurate comparison of expression profiles, GPC2 and SDC3 were analysed on the same blot. This approach ensures that any differences observed are attributable to biological variation rather than differences in blotting or detection conditions. Co-detection also enabled internal validation, confirming that each antibody produced discrete banding patterns in line with their known isoforms and processing fragments.

We appreciate the opportunity to clarify this point and have ensured that the revised manuscript now clearly distinguishes between HS chain-based detection (used for general GAG assessment) and core protein-specific Western blotting (used for specific proteoglycan quantification).

Reviewer 2 Report

Comments and Suggestions for Authors

This is a long paper on the behaviour of three human-derived neuronal cell lines: SH-SY5Y human neuroblastoma cell line; ReNcell VM, ventral mesencephalon-derived; and ReNcell CX, frontal cortex-derived. SH-SY5Y neuronal inductive cultures displayed a neuronal phenotype, evident from the cellular morphology, calcium activity, along with the  gene/protein expression of neuronal markers. ReNcell CX in inductive cultures has committed to the astrocyte lineage, yet remains immature, with longer time in culture likely needed to produce a more mature, homogenous population of astrocytes. ReNcell VM differentiated into a heterogeneous mixture of different neural cell populations, expressing lineage fate markers of both neuronal and glial lineages.

The heparan sulfate proteoglycans (HSPGs) produced by these cells were localized by immunofluorescent microscopy and Western Blotting, but were not further characterized.

The authors have shown differences in the expression of NDST2 , NDST4 (N-sulfotransferases-2 and 4), and HS2ST1 (heparan sulfate 2 sulfotransferase-1) genes, and concluded that there were differences in heparan sulfate N- and 6-O-sulfation patterns, although this point has not been demonstrated.

The addition of PDGF to ReNcell VM neural inductive cultures appeared to suppress the astrocyte lineage and prompt oligodendrocyte lineages. The expression of HSPGs did not change, but HS6ST1 and HPSE genes were upregulated, suggesting differences in HS structures. But the authors did not demonstrate this either. In subsequent lines (849-860) the authors continue to speculate on HS structure, based only on the expression of enzymes. It seems that they do not have the experimental basis to draw those conclusions.

Furthermore, the authors should pay closer attention to the international literature. For instance, two important groups in Brazil have been studying heparan sulfate proteoglycans for over 50 years, and have published hundreds of papers on these compounds (Nader HB, Dietrich CP, Pavão MS, among others). Curiously, none of these studies was cited in this paper. Other researchers, such as Esko JD and Iozzo RV, who published almost 200 papers on heparan sulfate proteoglycans were also not cited. One of the pioneering works on heparan sulfate proteoglycans in the nervous system, published in 1989 by Schubert D, was also not cited. In that work, the author says that “The multiple species of HSPGs within the nervous system have a variety of functions. These include roles in ECM assembly and a contribution from synaptic vesicles to synaptic cleft matrix”.

In summary, it seems that the authors should pay closer attention to the international literature, and it also seems that they do not have experimental basis to draw their conclusions. Further characterization of HSPGs, particularly the structure of HS chains, is necessary.

Author Response

Reviewer 2                                                                                                                                                                                            

This is a long paper on the behaviour of three human-derived neuronal cell lines: SH-SY5Y human neuroblastoma cell line; ReNcell VM, ventral mesencephalon-derived; and ReNcell CX, frontal cortex-derived. SH-SY5Y neuronal inductive cultures displayed a neuronal phenotype, evident from the cellular morphology, calcium activity, along with the gene/protein expression of neuronal markers. ReNcell CX in inductive cultures has committed to the astrocyte lineage, yet remains immature, with longer time in culture likely needed to produce a more mature, homogenous population of astrocytes. ReNcell VM differentiated into a heterogeneous mixture of different neural cell populations, expressing lineage fate markers of both neuronal and glial lineages.

The heparan sulfate proteoglycans (HSPGs) produced by these cells were localized by immunofluorescent microscopy and Western Blotting but were not further characterized.

The authors have shown differences in the expression of NDST2, NDST4 (N-sulfotransferases-2 and 4), and HS2ST1 (heparan sulfate 2 sulfotransferase-1) genes, and concluded that there were differences in heparan sulfate N- and 6-O-sulfation patterns, although this point has not been demonstrated.

The addition of PDGF to ReNcell VM neural inductive cultures appeared to suppress the astrocyte lineage and prompt oligodendrocyte lineages. The expression of HSPGs did not change, but HS6ST1 and HPSE genes were upregulated, suggesting differences in HS structures. But the authors did not demonstrate this either. In subsequent lines (849-860) the authors continue to speculate on HS structure, based only on the expression of enzymes. It seems that they do not have the experimental basis to draw those conclusions.

We thank the Reviewer for these comments and the opportunity to more fully clarify these queries. We agree that the manuscript some of the results are highly speculative and we do believe HS chain synthesis and sulfation enzymes likely play a significant role in mediating lineage specification in the models presented. However, mechanistic validation would require targeted functional studies (e.g., gene knockdown or enzymatic inhibition), which are beyond the scope of this characterisation-focused manuscript. This was also raised by another Reviewer. We have now clarified this in the Discussion and highlighted the value of our data as a foundation for future studies to explore the functional relevance of specific HS biosynthetic enzymes and modifications during neural differentiation. Specifically, we have now included the following (lines 881-889):

“While changes in HS sulfation as reflected by the differential expression of biosynthesis and modification enzymes may play a key role in HSPG-mediated lineage specification, it is essential to validate these findings through functional studies, such as targeted knockdown and enzymatic inhibition. Additionally, employing GAG sequencing techniques would be invaluable to characterise the temporal alterations in HS side chains during both in vitro and in vivo neural cell culture differentiation. We encourage future studies to pursue these approaches to further elucidate the mechanistic roles of HS modifications in neural lineage specification.”

Furthermore, the authors should pay closer attention to the international literature. For instance, two important groups in Brazil have been studying heparan sulfate proteoglycans for over 50 years, and have published hundreds of papers on these compounds (Nader HB, Dietrich CP, Pavão MS, among others). Curiously, none of these studies was cited in this paper. Other researchers, such as Esko JD and Iozzo RV, who published almost 200 papers on heparan sulfate proteoglycans were also not cited. One of the pioneering works on heparan sulfate proteoglycans in the nervous system, published in 1989 by Schubert D, was also not cited. In that work, the author says that “The multiple species of HSPGs within the nervous system have a variety of functions. These include roles in ECM assembly and a contribution from synaptic vesicles to synaptic cleft matrix”.

In summary, it seems that the authors should pay closer attention to the international literature, and it also seems that they do not have experimental basis to draw their conclusions. Further characterization of HSPGs, particularly the structure of HS chains, is necessary.

We thank the Reviewer for identifying additional literature of relevance that support this study, many of which were foundational contributions to the field. We acknowledge that our reference list was weighted toward recent and methodologically aligned studies, however we have now incorporated citations to key studies at lines 66 (Lopes, Dietrich, Nader), 68 (Esko) and 70 (Schubert) during the paper’s  introduction of HSPGs.

We also agree that further structural characterisation of HS chains would be an important next step to deepen our understanding of their specific roles during neural lineage specification (as per our previous response). However, while the current study focuses on gene and protein expression profiles as a foundation, we acknowledge this limitation and have now emphasised the need for future studies using approaches such as GAG sequencing and structural analysis to provide this insight.

Round 2

Reviewer 1 Report

Comments and Suggestions for Authors

The materials added to the supplemental file as part of the response were not available to this reviewer.  

Pg 7, line 297, first word should be BDNF?  This is a revised section, and likely a typo.

Author Response

The materials added to the supplemental file as part of the response were not available to this reviewer.  

We appreciate the Reviewer highlighting this issue and apologise for the inconvenience. The supplemental materials were indeed submitted alongside our revised manuscript; however, there might have been an issue with file accessibility through the review system. In brief, as suggested, we have now included a supplementary summary table of all primer sequences. We will contact the editorial office to ensure the supplemental materials are accessible. Additionally, we can directly provide these materials upon request to facilitate the review process. We again apologise for any inconvenience this may have caused.

Pg 7, line 297, first word should be BDNF?  This is a revised section, and likely a typo.

We thank the Reviewer for identifying this error. This was indeed a typographical error and has now been corrected. We have carefully revised the text on page 7, line 297 to correct the typo as follows:

"PDNF → BDNF"

The revised sentence now reads clearly and accurately:

"Western blot (WB) analysis was conducted using biological triplicates to assess HSPG core protein expression during ReNcell VM differentiation in the presence of BDNF and PDGF"

We appreciate this detailed feedback, which has allowed us to improve the manuscript’s accuracy and readability and have also made an effort to proof the entire manuscript.

Reviewer 2 Report

Comments and Suggestions for Authors

The title was modified, and it is much better now.

Only a few modifications were introduced in the text. Although the authors introduced a few new references and mentioned that further studies are necessary to support their conclusions, they have not yet clarified the limitations of their studies.

There is extensive data in the literature demonstrating that there can be significant differences between “enzyme expression” and “enzyme activity”. Many steps can affect this process: mRNA turnover, enzyme translation and processing speed, enzyme turnover, HSPG turnover, and so on. Therefore, concluding from enzyme expression that the HSPG structure is different is largely speculative.

I understand that it would be impossible, in this work, to carry out studies on the structure of the HS, but the authors should make the limitations of their work clearer.

Author Response

The title was modified, and it is much better now.

Only a few modifications were introduced in the text. Although the authors introduced a few new references and mentioned that further studies are necessary to support their conclusions, they have not yet clarified the limitations of their studies.

There is extensive data in the literature demonstrating that there can be significant differences between “enzyme expression” and “enzyme activity”. Many steps can affect this process: mRNA turnover, enzyme translation and processing speed, enzyme turnover, HSPG turnover, and so on. Therefore, concluding from enzyme expression that the HSPG structure is different is largely speculative.

I understand that it would be impossible, in this work, to carry out studies on the structure of the HS, but the authors should make the limitations of their work clearer.

We thank the Reviewer for their appreciation of the revised manuscript title and their insightful comments and acknowledge the importance of clarifying the limitations of our study. We fully agree with the Reviewer’s concern regarding the distinction between enzyme expression levels and their functional activity, as well as the speculative nature of correlating enzyme expression directly with changes in heparan sulfate (HS) structure. Accordingly, we have now revised the manuscript to explicitly clarify these limitations.

Revised Manuscript Text (Limitations Section added at the end of the Discussion section): Line 922-949

  1. Limitations

While this study provides valuable insights into the expression profiles of HSPGs and related biosynthetic enzymes during neural lineage specification, several limitations must be acknowledged. Firstly, our findings primarily rely on gene and protein expression analyses, which do not necessarily reflect enzyme or protein functional activity. It is well-documented that enzyme activity can significantly differ from expression levels due to various biological processes, such as differences in mRNA stability, translational efficiency, post-translational modifications, protein stability, and enzyme turnover rates [96–99]. Thus, caution must be taken when interpreting our data in terms of functional outcomes.

Secondly, our study did not include direct biochemical analyses of HS GAG structures or sulfation patterns. Such analyses are crucial for confirming functional implications suggested by changes in gene and protein expression. Variability in HS structure, including sulfation patterns, chain length, and epimerisation, necessitates detailed structural analysis using methods such as mass spectrometry or high-performance liquid chromatography (HPLC). These techniques were beyond the scope of the present investigation [100,101]. Consequently, interpretations linking observed expression changes directly to alterations in HS GAG structure should be considered tentative.

Furthermore, the use of immortalised neural cell lines (SH-SY5Y, ReNcell CX, and ReNcell VM), while beneficial for reproducibility and ease of handling, presents inherent limitations including reduced physiological relevance when compared to primary neural progenitors with some potential for culture-induced artifacts [102,103]. Thus, generalising our findings to primary neural progenitors or in vivo conditions must also be approached cautiously.  Further validation approaches using primary or stem-cell-derived neural cells would be required to enhance physiological relevance. As an example, future investigations could incorporate direct functional assays, detailed HS structural analyses, and comparative studies with primary neural cells to validate and extend the findings presented in this study.

References

  1. Kowarz E, Löscher D, Marschalek R. Optimized Sleeping Beauty transposons rapidly generate stable transgenic cell lines. Biotechnol J. 2015;10(4):647-53.
  2. Horvathova I, Voigt F, Kotrys AV, Zhan Y, Artus-Revel CG, Eglinger J, et al. The Dynamics of mRNA Turnover Revealed by Single-Molecule Imaging in Single Cells. Mol Cell. 2017;68(3):615-25.e9.
  3. Hinkson IV, Elias JE. The dynamic state of protein turnover: It's about time. Trends Cell Biol. 2011;21(5):293-303.
  4. Christianson HC, Belting M. Heparan sulfate proteoglycan as a cell-surface endocytosis receptor. Matrix Biol. 2014;35:51-5.
  5. Esko JD, Selleck SB. Order out of chaos: assembly of ligand binding sites in heparan sulfate. Annu Rev Biochem. 2002;71:435-71.
  6. Annaval T, Wild R, Crétinon Y, Sadir R, Vivès RR, Lortat-Jacob H. Heparan Sulfate Proteoglycans Biosynthesis and Post Synthesis Mechanisms Combine Few Enzymes and Few Core Proteins to Generate Extensive Structural and Functional Diversity. Molecules. 2020;25(18).
  7. Azari H, Rahman M, Sharififar S, Reynolds BA. Isolation and expansion of the adult mouse neural stem cells using the neurosphere assay. J Vis Exp. 2010(45).
  8. Donato R, Miljan EA, Hines SJ, Aouabdi S, Pollock K, Patel S, et al. Differential development of neuronal physiological responsiveness in two human neural stem cell lines. BMC Neurosci. 2007;8:36.

Round 3

Reviewer 2 Report

Comments and Suggestions for Authors

The limitations of this study were clearly stated. The paper can be published in its present form.